# BENCHMARKING RELIABILITY AND GENERALIZATION BEYOND CLASSIFICATION

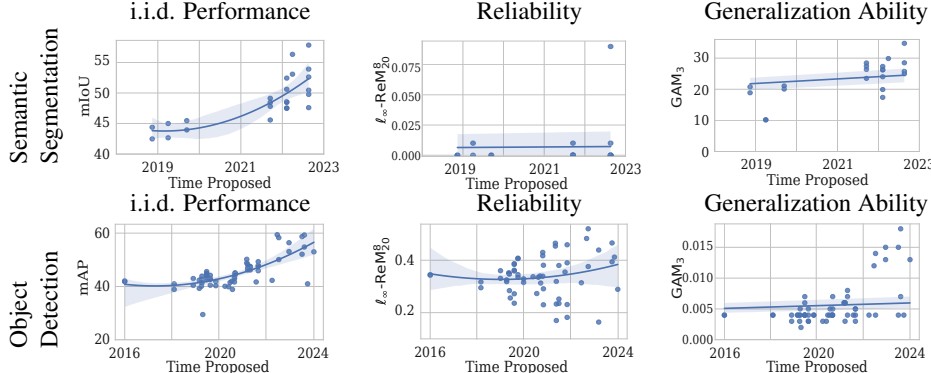

Figure 1: An overview of semantic segmentation (TOP) and object detection (BOTTOM) methods proposed over time and their reliability and generalization ability on ADE20K (Zhou et al., 2019) and MS-COCO (Lin et al., 2014), respectively. The y-axes represent TOP: the mean Intersection over Union (mIoU) and BOTTOM: the mean Average Precision (mAP), i.e. higher is better. The performance of methods on i.i.d. samples has increased over time, however, their reliability and generalization ability have not improved at the same rate, and lag behind.

## ABSTRACT

Reliability and generalization in deep learning are predominantly studied in the context of image classification. Yet, possible real-world applications in safety-critical domains involve a broader set of semantic tasks, such as semantic segmentation and object detection, which come with a diverse set of dedicated model architectures. To facilitate research towards robust model design in segmentation and detection, our primary objective is to provide benchmarking tools regarding robustness to distribution shifts and adversarial manipulations. We propose the benchmarking tools SEMSEGBENCH and DETECBENCH, along with the most extensive evaluation to date on the reliability and generalization of semantic segmentation and object detection models. Specifically, we benchmark 76 segmentation models across four datasets and 61 object detectors across two datasets, evaluating their performance under diverse adversarial attacks and common corruptions. Our findings reveal systematic weaknesses in state-of-the-art models and uncover key trends based on architecture, backbone, and model capacity. SEMSEGBENCH and DETECBENCH are open-sourced in an Anonymous Repository with our complete set of 6139 evaluations. We anticipate the collected data to foster and encourage future research towards improved model reliability beyond classification.

## 1 INTRODUCTION

Deep Learning (DL)-based models can provide highly accurate predictions. At the same time, they are known to behave unstably under distribution shifts (Hendrycks & Dietterich, 2019; Kar et al., 2022; Hooker et al., 2019) or when probed using attacks (Kurakin et al., 2017; Croce & Hein, 2020; Schmalfuss et al., 2022a). This known drawback casts serious doubts over their use for safety-critical applications such as medical image analysis (Sobek et al., 2024; Ran et al., 2023; Ronneberger et al., 2015; Dumitru et al., 2023) or autonomous driving (Balasubramaniam & Pasricha, 2022; Khan et al.,

2023; Menze & Geiger, 2015; Cordts et al., 2016). Most prior works (Geirhos et al., 2018; Prasad, 2022; Geirhos et al., 2020; Liu et al., 2025) studied these shortcomings for image classification models, yielding large studies and benchmarks (Croce et al., 2021; Hendrycks & Dietterich, 2019; Kar et al., 2022; Jung et al., 2023) that foster research towards improved model reliability and robustness. However, this incentive hardly transfers from classification models to models for semantic segmentation and object detection. At the same time, findings from image classification might not directly transfer to e.g. segmentation models due to systematic differences in architecture design and loss, while the role of model design was highlighted e.g. in (Schrodi et al., 2022; Agnihotri et al., 2023b; 2024a) who exemplified architectural design choices that improve model robustness independent of training strategies.

Most current segmentation and detection models are evaluated on and optimized for performance on independent and identically distributed (i.i.d.) in-domain samples. Arguably, evaluating recent state-of-the-art models also incurs relevant compute costs. Therefore, while highly relevant in practice, no large-scale studies exist regarding model robustness and reliability for semantic segmentation and object detection, potentially stagnating progress towards improved robustness and reliability. This is observed in Fig. 1 for both semantic segmentation and object detection methods over time.

To alleviate this gap, we perform a comprehensive benchmarking of semantic segmentation and object detection methods for reliability under adversarial attacks and generalization to 2D and 3D common corruptions. Additionally, to ensure smooth, continued work in this direction, we propose two new benchmarking tools, SEMSEGBENCH and DETECBENCH for semantic segmentation and object detection, respectively. SEMSEGBENCH is built upon "mmsegmentation" (Contributors, 2020) and DETECBENCH is built upon "mmdetection" (Chen et al., 2019). This allows our proposed benchmarking tools to cover most relevant DL-based methods for the tasks and to be easily updated over time as methods are added to "mmsegmentation" and "mmdetection" while getting community-trusted documentation.

The proposed SEMSEGBENCH and DETECBENCH are the first unified benchmarking tools for evaluating the adversarial and OOD robustness of their respective downstream tasks. These benchmarks uniquely enable joint analysis of semantic segmentation and object detection, leveraging their common architectural backbones and vulnerabilities. With **6,139 evaluations** covering **76 segmentation models** and **61 object detectors**, the extensive benchmark provides multiple pre-logged metrics per evaluation, allowing the community to immediately explore further analyses without computation positioning SEMSEGBENCH and DETECBENCH as not just tools, but comprehensive resources for research towards reliable models.

Our analysis provides a novel understanding of various methods across datasets and allows to identify trends for model reliability and generalization. Thus, we anticipate insights from this work to help researchers build better models that are not limited to improved performance on i.i.d. samples but are additionally less vulnerable to adversarial attacks while generalizing better to image corruptions, ultimately allowing for their safer deployment in practice. Our main contributions are:

- For semantic segmentation, we evaluate 76 checkpoints over 4 datasets, 3 SotA adversarial attacks, and 15 2D Common Corruptions with a total of 2052 evaluations.

- For object detection, we evaluate 61 methods across 2 different datasets under 3 diverse adversarial attacks and 25 established common corruptions with a total of 4087 evaluations.

- We perform the most comprehensive analysis of semantic tasks beyond classification to date, analyzing correlations between performance, reliability, and the generalization abilities of semantic segmentation and object detection methods under various architectural design choices, such as model capacity or the type of model backbone used.

- We empirically show that synthetic image corruptions can serve as a proxy for real-world distribution shifts (included as Appendix A.1 due to page limit).

- We propose SEMSEGBENCH and DETECBENCH, the first unified benchmarking tools for in-distribution performance, OOD, and adversarial robustness of most DL-based methods over the most commonly used datasets for the respective tasks.

- New summary metrics: Reliability Measure (ReM) and Generalization Ability Measure (GAM), which condense robustness results into interpretable worst-case evaluations across corruptions and attacks.

## 2 RELATED WORK

Previous works have benchmarked either OOD robustness (Kamann & Rother, 2020; Gupta et al., 2024; Michaelis et al., 2019; Ding et al., 2024) or adversarial robustness (Arnab et al., 2018; Croce et al., 2024; Awais et al., 2023; Dong et al., 2022; Xie et al., 2017a; Du et al., 2022) for segmentation and object detection. SEMSEGBENCH and DETECBENCH are the first to unify these evaluations for the respective tasks, inspired by benchmarks for image classification (Croce et al., 2021).

**Adversarial Attacks**   DL-based models tend to rely on unreliable features (Geirhos et al., 2020), which makes them susceptible to adversarial attacks (Goodfellow et al., 2015) that exploit these vulnerabilities and have thus become a community-accepted protocol for measuring model reliability. Early methods like FGSM (Goodfellow et al., 2015) led to more advanced attacks such as BIM (Kurakin et al., 2018), PGD (Kurakin et al., 2017), and CosPGD (Agnihotri et al., 2024b). Attacks like SegPGD (Gu et al., 2022) and CosPGD are specialized for semantic segmentation, and are thus used in SEMSEGBENCH. For DETECBENCH, we use generic attacks (BIM, PGD) for consistency across architectures as the predominant use of generic attacks allows a more generic framework. This does not depreciate the importance of the benchmark's findings since we are interested in the relative performance of the methods and not the absolute performance. Moreover, extending prior object detection attacks (Xie et al., 2017a; Wei et al., 2019) to all architectures is not straightforward and might need adapting them to specific architectures.

**OOD Robustness**   Deep learning models often fail under distribution shifts, making OOD robustness a critical measure of generalization (Hendrycks et al., 2020; Hoffmann et al., 2021). For semantic segmentation, OOD robustness has been evaluated using synthetic corruptions like Common Corruptions (Hendrycks & Dietterich, 2019) and 3D Common Corruptions (Kar et al., 2022), as well as real-world conditions in ACDC (Sakaridis et al., 2021). For object detection, prior works focus on weather shifts (Gupta et al., 2024), challenging environments (Ding et al., 2024), and corruption benchmarks (Michaelis et al., 2019). Our SEMSEGBENCH extends these by using both synthetic and real-world corruptions for semantic segmentation, while DETECBENCH uses synthetic corruptions exclusively, providing a unified framework to assess OOD robustness as a measure of generalization.

**Robustness Benchmarking Tools**   RobustBench (Croce et al., 2021) and RobustArts (Tang et al., 2021) are popular robustness benchmarks for image classification. While several works have benchmarked OOD robustness for semantic segmentation (Kamann & Rother, 2020; Sakaridis et al., 2021) and object detection (Gupta et al., 2024; Michaelis et al., 2019), their evaluations are often limited to specific architectures, datasets, or corruption types. For adversarial robustness, existing tools like Torchattacks (Kim, 2020) and Foolbox (Rauber et al., 2020) focus on classification. (Chan et al., 2021) provides a benchmark for segmentation model robustness to anomalies. Complementing the above works, SEMSEGBENCH and DETECBENCH provide the first unified framework for large-scale evaluation of both OOD and adversarial robustness for semantic segmentation and object detection, covering a diverse set of architectures and datasets.

## 3 METRICS FOR ANALYSIS AT SCALE

This work is the first to analyze semantic segmentation and object detection under the lens of reliability and generalization at such a large scale.

For semantic segmentation, we use the standard metrics: mean Intersection over Union (mIoU), mean class Accuracy (mAcc), and mean pixel Accuracy (aAcc) (Zhao et al., 2017; Arnab et al., 2018; Agnihotri et al., 2024b). In the Appendix, we show a strong correlation between these metrics.

For object detection, we follow standard practice (Ren, 2015; Lin, 2017; He et al., 2017; Cai & Vasconcelos, 2019; Li et al., 2022b) and use mean Average Precision (mAP). For MS-COCO, we also report $mAP_{small}$, $mAP_{medium}$, and $mAP_{large}$, while for PASCAL VOC, we include mAP, mAP25, mAP50, and mAP75. Correlation among these metrics is detailed in the Appendix.

Given the extensive evaluations using SEMSEGBENCH and DETECBENCH across diverse corruptions and attacks, we introduce two metrics to simplify analysis: *Reliability Measure* and *Generalization Measure*. These capture the worst-case performance (mIoU for segmentation, mAP for detection) across all corruptions and attacks on a dataset, answering the question: "What is the

worst-case performance of a method?" Our findings remain consistent when using average values, as shown in the Appendix.

### 3.1 Reliability Measure

As discussed by (Agnihotri et al., 2024b; Gu et al., 2022; Schmalfuss et al., 2022b), white box adversarial attacks serve as a proxy for the worst-case scenario for a method, and thus a method's performance against such attacks serves as a viable measure of its reliability. Thus, we propose using ReliabilityMeasure (ReM) as the measure of performance against adversarial attacks. Here, we consider iterative non-targeted adversarial attacks. For SEMSEGBENCH, we use PGD, SegPGD, and CosPGD and calculate the mIoU of the predicted segmentation mask under attack w.r.t. the ground truth segmentation mask. For DETECBENCH, we use BIM and PGD and calculate the mAP of the predicted bounding boxes under attack w.r.t. the ground truth bounding boxes. A higher ReM value indicates better reliability. Some prior works (Xie et al., 2017a; Wei et al., 2019; Jia et al., 2022; Cai et al., 2022; Huang et al., 2023) have proposed adversarial attacks, especially transfer attacks, specifically for some object detection methods. However, the goal of this work is to measure the reliability and generalization ability of methods as a whole. Thus, for fairness in evaluations, we need attacks that work across all model architectures without the need for adapting the attack. Thus, we focus on generic and widely used vision attacks like BIM and PGD in DETECBENCH.

Constrained adversarial attacks can be optimized under various $\ell_p$-norms. We focus on the two most commonly used (Agnihotri et al., 2024b; Kurakin et al., 2017; Madry et al., 2017; Wong et al., 2020; Schmalfuss et al., 2022b) $\ell_p$-norms, i.e. $\ell_\infty$-norm and $\ell_2$-norm. The notation for this metric is, $\ell_p$-$\mathrm{ReM}^\epsilon_{attack\ iterations}$, where the subscript informs the number of attack iterations used for optimizing the attack, and the superscript is the permissible perturbation budget $\epsilon$. For example, when 20 attack iterations were used to optimize an $\ell_\infty$-norm bounded attack with $\epsilon = \frac{8}{255}$ under $\ell_\infty$-norm constrain then the metric would be $\ell_\infty$-$\mathrm{ReM}^8_{20}$. We limit the analysis to 20 attack iterations since most previous works (Gu et al., 2022; Schmalfuss et al., 2022b; Agnihotri et al., 2024b; 2023a) on adversarial robustness for various tasks have shown that 20 attack iterations is enough optimization budget for attacks, especially when reporting relative performance of methods.

### 3.2 Generalization Ability Measure

Multiple image classification works (Croce et al., 2021; Hendrycks et al., 2020; Hoffmann et al., 2021) and some semantic segmentation (Kamann & Rother, 2020) and Object Detection (Gupta et al., 2024; Michaelis et al., 2019) works use OOD Robustness of methods for evaluating the generalization ability of the method, however, different image corruptions impact the performance of the semantic segmentation methods differently. As we are interested in the worst possible case, we define GeneralizationAbilityMeasure (GAM), as the worst mIoU (SEMSEGBENCH) or worst mAP (DETECBENCH) across all image corruptions at a given severity level. We find the minimum of the mIoU of the segmentation masks predicted under image corruptions w.r.t. the ground truth masks for a given method, across all corruptions at a given severity and report this as the $\mathrm{GAM}_{severity\ level}$. For example, for severity=3, the measure would be denoted by $\mathrm{GAM}_3$. The higher the GAM value, the better the generalization ability of the given semantic segmentation method.

The GAM value is calculated over the following 2D Common Corruptions (Hendrycks & Dietterich, 2019): 'gaussian noise', 'shot noise', 'impulse noise', 'defocus blur', 'frosted glass blur', 'motion blur', 'zoom blur', 'snow', 'frost', 'fog', 'brightness', 'contrast', 'elastic', 'pixelate', 'jpeg'. Additionally, in DETECBENCH, we also evaluate the following 3D Common Corruptions (Kar et al., 2022): 'color quant', 'far focus', 'fog 3d', 'iso noise', 'low light', 'near focus', 'xy motion blur', and 'z motion blur'. Please refer to the Appendix for mean performance over all corruptions.

## 4 Analysis And Key Findings

### 4.1 Semantic Segmentation

SEMSEGBENCH supports all semantic segmentation methods included in mmsegmentation, multiple $\ell_p$-norms for adversarial attacks, and all severity levels for common corruptions. However, using

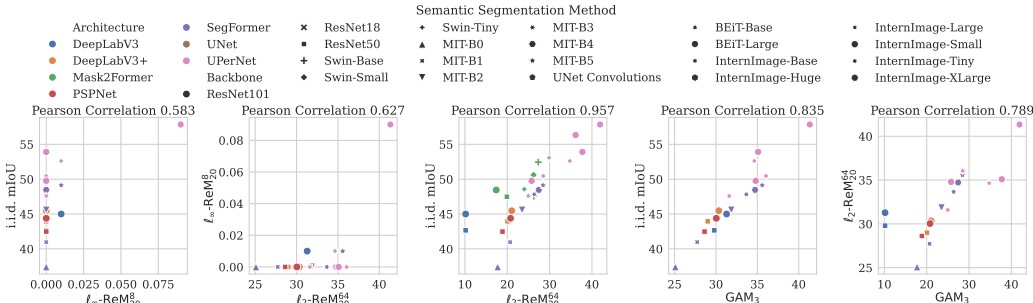

(a) Correlations between (1) i.i.d. performance and reliability under $\ell_\infty$-norm attacks, (2) reliability under $\ell_\infty$-norm and $\ell_2$-norm attacks, (3) i.i.d. performance and reliability under $\ell_2$-norm attacks, (4) i.i.d. performance and generalization ability, and (5) reliability under $\ell_2$-norm attacks and generalization ability. See Appendix for further analysis.

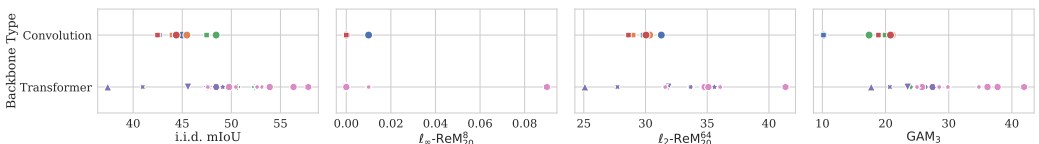

(b) Analysis of i.i.d. performance, reliability, and generalization based on backbone type (CNN-based at top, vision transformer-based at bottom). Marker colors indicate method architecture, and shapes denote backbones. From left to right, x-axis shows: (1) i.i.d. performance, (2) reliability under $\ell_\infty$-norm attacks, (3) reliability under $\ell_2$-norm attacks, and (4) generalization ability. Legend as in Fig. 2a.

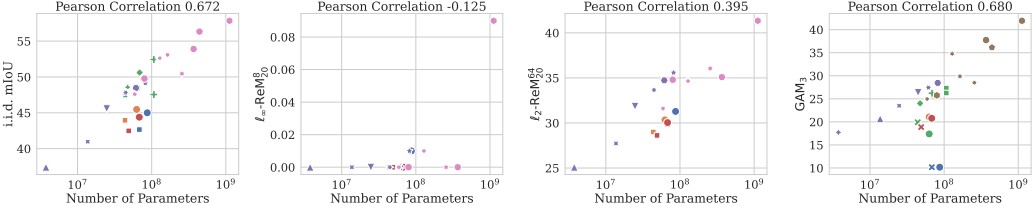

(c) Correlation of i.i.d. performance, reliability, and generalization with the number of learnable parameters (log scale). From left to right: (1) i.i.d. performance, (2) reliability under $\ell_\infty$-norm attacks, (3) reliability under $\ell_2$-norm attacks, and (4) generalization ability. Legend as in Fig. 2a.

Figure 2: **Semantic Segmentation** using the ADE20K dataset. The colors represent the architecture of the method, while marker shapes represent the backbone of the respective method. All methods were trained on the train set of ADE20K. Please refer to the Appendix for results with other datasets i.e. Cityscapes and PASCAL VOC2012. Additionally, in the Appendix, we show high correlation between performance across different datasets. Subfigure numbers are left to right.

all the methods and performing all possible evaluations is not realistic due to it being computationally very expensive (e.g. training all missing model checkpoints or running all possible combinations of attacks). Thus, for this benchmark, we use the most prominent semantic segmentation methods proposed and used over the years. As most semantic segmentation works show that the model backbone has a huge impact on the performance of the method, we attempt to include a broad spectrum of backbones in the analysis. Please refer to the Appendix for dataset details, additional implementation details, and additional results (detailed for each threat model) from the benchmarking.

### 4.1.1 PERFORMANCE V/S RELIABILITY V/S GENERALIZATION

As discussed in Sec. 1, semantic segmentation methods proposed over the years have improved their performance on i.i.d. samples. Using SEMSEGBENCH we can now provide a large-scale analysis for correlations between the performance and reliability, and generalization abilities of semantic segmentation models First, we can confirm observations by (Gu et al., 2022; Agnihotri et al., 2024b) that semantic segmentation methods are not inherently robust to strong adversarial attacks, especially those bounded by the $\ell_\infty$-norm.

Second, we also observe that typical $\ell_\infty$-norm attacks with $\epsilon = \frac{8}{255}$ as used by (Agnihotri et al., 2024b; Gu et al., 2022) are simply too strong to analyze correlations. They cause almost all semantic segmentation methods to completely fail. This raises credible concerns regarding the reliability of semantic segmentation methods under $\ell_\infty$-norm attacks. Some methods like (Xu et al., 2021; Croce et al., 2024) attempt to address this, please refer to the appendix for evaluations using these. However, semantic segmentation methods seem slightly more robust to $\ell_2$-norm bounded adversarial attacks. **For reliability under $\ell_2$-norm attacks, we find that there does exist a strong correlation between i.i.d. performance and reliability.**

Next, we find some correlation between i.i.d. performance and generalization ability. Some methods like InternImage achieve both, descent i.i.d. performance and OOD robustness. However, methods such as DeepLabV3 (Chen et al., 2017) do not appear to have a good generalization ability. Please note that even InternImage, while generalizing better than the other models, still lacks true generalization abilities, which would ensure a minimal drop in performance between i.i.d. and OOD samples. However, we see a significant drop for InternImage. Additionally, we observe that while SegFormer (Xie et al., 2021) and InternImage (Wang et al., 2023b) are not robust against $\ell_\infty$-norm attacks, they are robust to some extent against $\ell_2$-norm attack when using large backbones (relatively higher number of parameters). This aligns with the observations made for image classification models, where models with a large number of parameters were found to be more robust (Croce et al., 2021) and will be subject to discussion in the following subsection.

For reliability and generalization ability, we observe a positive, but not strong, correlation between the two only when using $\ell_2$-norm as the reliability measure. **Thus, future methods addressing one aspect, reliability (especially under $\ell_\infty$-norm attacks) or OOD generalization, might not inherently address the other, and each aspect requires intentional focus.**

### 4.1.2 ANALYZING THE BACKBONE TYPE

The feature extractor used by a semantic segmentation method significantly impacts its performance. Common backbones are primarily of two types, Convolution Neural Network (CNN) based (He et al., 2016; Liu et al., 2022b; Chen et al., 2017; Zhao et al., 2017; Agnihotri et al., 2023b), or are Vision Transformer based (Liu et al., 2021; Wang et al., 2023b; Xie et al., 2021). Thus, to better understand the reliability and generalization abilities of semantic segmentation methods, we analyze them based on the backbone type used by them in Fig. 2b. Please note that a given method comprises an architecture and a backbone, and each architecture can be coupled with different backbones.

We observe in Fig. 2b that **models with transformer-based backbones have significantly better generalization abilities.** Previous works (Paul & Chen, 2022; Hoyer et al., 2022; Xie et al., 2021) explain this by showing that transformer-based models are inherently more OOD robust than CNN-based models. Lastly, we observe that transformer-based models also have slightly better i.i.d. performance, this can especially be seen using Mask2Former, which performs better with transformer-based backbones than with CNN-based backbones.

### 4.2 OBJECT DETECTION

DETECBENCH supports all object detection methods provided by *mmdetection*. However, with similar reasons as for SEMSEGBENCH, we restrict our evaluation to a practically feasible setting: We identify and benchmark the most prominent object detection methods proposed over the years, including some recent SotA methods. As for segmentation, previous works (Zong et al., 2023; Zhang et al., 2023; Li et al., 2022c; Lin, 2017; Ren, 2015; Li et al., 2022b; Cai & Vasconcelos, 2019; Zhang et al., 2020b) highlight the importance of the feature extraction backbone for model performance. We account for this fact in our analysis. We focus our experiments on the MS-COCO dataset (Lin et al., 2014). Please refer to the Appendix for details on the dataset, additional implementation details, and additional results (detailed for each threat model).

### 4.2.1 PERFORMANCE V/S RELIABILITY V/S GENERALIZATION

In Fig. 3a, we first analyze correlations between the object detection methods' reliability and generalization ability. We also look for a correlation between both separately with i.i.d. performance.

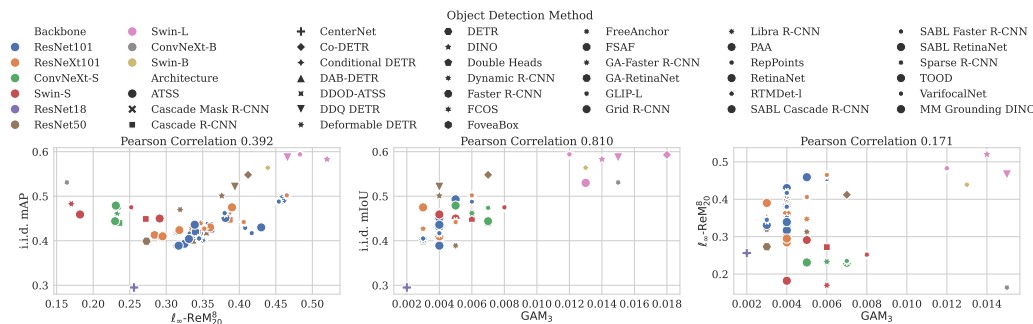

(a) Correlations between (1) i.i.d. performance and reliability under $\ell_\infty$-norm attacks, (2) i.i.d. performance and generalization ability, and (3) reliability under $\ell_\infty$-norm attacks and generalization ability.

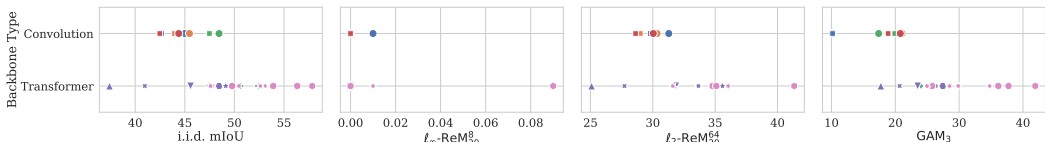

(b) Analysis of i.i.d. performance, reliability, and generalization ability based on backbone type (CNN-based at top, vision transformer-based at bottom). From left to right: (1) i.i.d. performance, (2) reliability under $\ell_\infty$-norm attacks, and (3) generalization ability. Legend as in Fig. 3a.

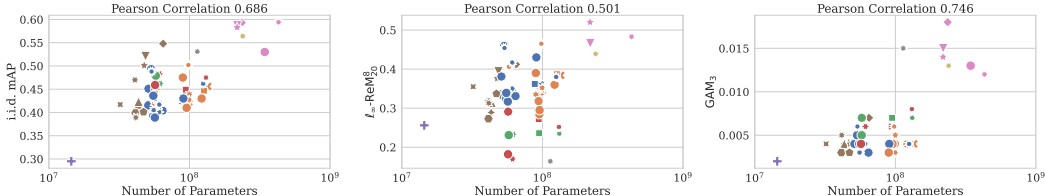

(c) Correlation of i.i.d. performance, reliability, and generalization ability with the number of learnable parameters (log scale). From left to right: (1) i.i.d. performance, (2) reliability under $\ell_\infty$-norm attacks, (3) reliability under $\ell_2$-norm attacks, and (4) generalization ability. Legend as in Fig. 3a.

Figure 3: **Object Detection** on MS-COCO. The colors represent the backbone of the respective method, while marker shapes represent the architecture of the method. All methods were trained on the MS-COCO train set. The numbers in subcaptions for the respective subfigures are left to right.

First, when looking at the i.i.d. performance and reliability under adversarial attacks, we see a weak positive correlation. Few methods like MM Grounding DINO (Zhao et al., 2024), DDQ-DETR (Zhang et al., 2023), GLIP (Li et al., 2022c), DINO (Caron et al., 2021), and RTMDet (Lyu et al., 2022) stand out, however, only when using a large backbone like Swin-L (Liu et al., 2021), Swin-B (Liu et al., 2021), ResNet101 (He et al., 2016) or ResNeXt101 (Xie et al., 2017b). They show good i.i.d. performance and good reliability under adversarial attacks.

Second, we consider the correlation between i.i.d. performance and generalization ability to image corruptions. Again, we observe that DDQ-DETR, GLIP, RTMDet, and DINO show good i.i.d. performance and relatively better generalization ability, however, only when using a Swin-L, Swin-B, or ConvNeXt-B (Liu et al., 2022b) backbone. With other backbones such as ResNet50: DDQ-DETR, DINO, and Co-DETR have only descent i.i.d. performance and very poor OOD robustness. There is a strong positive correlation between i.i.d. performance and the generalization ability of object detection methods. This aligns with our findings for semantic segmentation. Please note that all considered object detection methods have a very low $\mathrm{GAM}_3$ value, which aggregates 2D and 3D common corruptions. We report the individual values in the Appendix.[1]

---

[1]We observe that for most corruptions, the values are reasonable, though still significantly lower than their i.i.d. performance. However, the methods seem to completely fail against a few 3D common corruptions (Kar et al., 2022) such as xy motion blur, z motion blur, and fog image corruptions. This brings down the resultant $\mathrm{GAM}_3$. Yet we argue for including these in the $\mathrm{GAM}_3$ values because these are very realistic. Moreover,

Third, Fig. 3a (right) correlates reliability and generalization ability. However, the low $\mathrm{GAM}_3$ scores hardly allow to draw any conclusions and we do not observe any correlation between reliability under attack and OOD generalization. Please note, given that Co-DETR and MM Grounding DINO exhibit the best relative generalization ability and i.i.d. performance, it would be interesting to assess their reliability under adversarial attacks, which however exceeds our computational resources.

From observations made on Fig. 3a, we conclude that there exists no correlation between performance and reliability, and reliability and generalizations, but there exists a strong correlation between performance and generalization. Since most works on object detection only focus on i.i.d. performance, this observation provides reasoning to the arguments made using Fig. 1 in Sec. 1. **Reliability and OOD generalization do not necessarily improve with improved i.i.d. model performance. Thus, to obtain reliable and OOD generalizable object detection, conscious effort needs to be focused in this direction.**

### 4.2.2 Analyzing Backbones Type

Next, we analyze the backbone design decision in more detail by dividing them into two broad categories, Convolution Neural Network (CNN) based backbones such as ResNets (He et al., 2016), ResNeXts (Xie et al., 2017b), and ConvNeXts (Liu et al., 2022b), and vision transformer based backbones such as Swin transformers (Liu et al., 2021) Swin-S, Swin-B and Swin-L. In Fig. 3b, we observe that i.i.d. performance and reliability against adversarial attacks are comparable for both the types of backbones, with methods with transformer-based backbones achieving marginally better values. However, when considering generalization to OOD samples, transformer-based backbones clearly outperform CNN-based backbones, with the exception of ConvNeXt-B, which achieves relatively similar generalization ability. From these observations, we draw the conclusion that **transformer-based backbones have a slight edge over CNN-based backbones when it comes to i.i.d. performance, reliability against adversarial attacks, and generalization ability to OOD samples**. This observation aligns with our findings for semantic segmentation in Sec. 4.1.2 and with (Paul & Chen, 2022; Hoyer et al., 2022; Xie et al., 2021).

### 4.3 Key Finding of the Joint Analysis

We observe that despite semantic segmentation and object detection both being semantic tasks that often employ similar backbone encoders, the reliability and generalization abilities of their methods have some contrasting behaviors. For semantic segmentation, we observe a strong positive correlation in i.i.d. performance, reliability under $\ell_2$-norm attacks, and generalization ability. However, for object detection, we do not see this correlation, only a strong positive correlation between i.i.d. performance and generalization ability. Our key findings are summarized in Tab. 1. In both applications, our analysis indicates that reliability and generalization ability of models do not always improve with i.i.d. performance, i.e. dedicated effort has to be invested to make models behave in a reliable way. Some of the observed tendencies align with the observations made for image classification; e.g., methods with vision transformer backbones exhibit relatively better generalization ability to OOD data for both semantic segmentation and object detection (Paul & Chen, 2022; Hoyer et al., 2022). Conversely, some observations do not align for both tasks and past observations made for classification (Hooker et al., 2019). One such often discussed example for image classification is the impact of the model capacity (the number of learnable parameters) (Hooker et al., 2019; Hoefler et al., 2021). As last point, we extend this analysis to semantic segmentation and object detection in SemSegBench and DetecBench, respectively.

**Impact Of The Number Of Learnable Parameters** For semantic segmentation (Fig. 2c), we see a moderate correlation between the number of learnable parameters and i.i.d. performance. However, the correlation with reliability and generalization is weaker. Notably, Mask2Former, with significantly fewer parameters than InternImage, shows higher robustness under $\ell_\infty$ and $\ell_2$-norm attacks, suggesting that simply increasing model size does not guarantee better reliability or generalization. However, this could be due to inherent gradient obfuscation introduced by its attention masking mechanism and merits further investigation.

---

these poor $\mathrm{GAM}_3$ values draw attention to an almost complete lack of generalization, even for methods like Co-DETR (Zong et al., 2023) and MM Grounding DINO (Zhao et al., 2024).

Table 1: Summary of key findings, made in our main paper and in the Appendix, of our analysis of model reliability and generalization on semantic tasks beyond classification.

| Findings | Semantic Segmentation | Object Detection |
|---|---|---|
| **Correlations between i.i.d. performance, reliability and generalization** | Strong positive correlation (except $\ell_\infty$-norm attacks) (Sec. 4.1.1) | Strong Positive correlation only between i.i.d. and Generalization to OOD, None otherwise (Sec. 4.2.1). |
| **Are transformer-based methods more OOD Robust?** | Yes (Sec. 4.1.2) | Yes (Sec. 4.2.2) |
| **Is increased model capacity better for reliability and generalization?** | Moderately (Sec. 4.3) | Moderately (Sec. 4.3) |
| **Unique Findings** | Strong Correlation between Real and Fake Corruptions (Appendix) | High Positive Correlation between 2D and 3D Common Corruptions (Appendix) |
| **Strong Positive Correlation Across Metrics** | Yes (Appendix) | Yes (Appendix) |

Similarly, in object detection (Fig. 3c), we also find a moderate correlation between model size and performance, reliability, and generalization. Methods like DDQ-DETR, DINO, and Co-DETR with larger backbones generally perform better, while smaller models like CenterNet with ResNet18 consistently perform worse. Yet, these trends are not absolute, and other factors, such as backbone type (Sec. 4.2.2) and proposal technique (Appendix), appear to have a greater impact.

Overall, our analysis across both tasks reveals that while the number of parameters has a moderate effect on performance, reliability, and generalization, this effect is far less pronounced than expected from image classification (Hooker et al., 2019). Architectural design choices, rather than mere model size, are more critical determinants of robustness in semantic segmentation and object detection.

## 5 CONCLUSION

We present SEMSEGBENCH and DETECBENCH, the most comprehensive benchmarking tools to date for rigorously evaluating the robustness and generalization of semantic segmentation and object detection methods. Our benchmarks cover a vast suite of experiments across multiple datasets, diverse corruption types, and adversarial attacks, establishing an unprecedented evaluation framework for in-depth analysis of these critical tasks. Our evaluation uncovers the strong impact of architectural design choices, such as backbone type (CNN vs. vision transformer) and parameter count, and reveals performance correlations across diverse metrics and datasets. Although the influence of design choices on robustness is not novel, our framework enables a systematic and empirical study of which decisions most affect model reliability under varying conditions. This breadth of analysis, extending to previously underexplored scenarios, offers a nuanced understanding of model behavior beyond standard metrics. While our study does not propose new robustness methods or attacks, it establishes a solid foundation for future work by providing a standardized, transparent, and scalable way to assess robustness. By offering this unified framework, we pave the way for the development of more reliable and generalizable models, facilitating their safer potential deployment.

**Future Work** We plan to extend SEMSEGBENCH and DETECBENCH with additional distribution shifts, including lens aberrations (Müller et al., 2023), enabling a broader evaluation of real-world robustness. Moreover, while our current evaluations focus on pretrained and adversarially trained models (Appendix), we aim to support benchmarking for adversarial training methods directly within the frameworks (Kurakin et al., 2017; Agnihotri et al., 2024b; Xu et al., 2021; Croce et al., 2024) and inference time defenses Grabinski et al. (2022); Zhang (2019); Zou et al. (2023); Li et al. (2020). We also plan to integrate 3D Common Corruptions (Kar et al., 2022) for more realistic OOD evaluations in SEMSEGBENCH, and explore object detection-specific adversarial attacks (Xie et al., 2017a; Wei et al., 2019) in DETECBENCH, providing a more comprehensive analysis.

**Limitations** Our benchmarks are computationally intensive, making exhaustive evaluation of all possible architectures, settings, and attack configurations infeasible. We prioritize a limited but diverse set of models and tasks to maintain a manageable scope. Additionally, SEMSEGBENCH includes both real-world and synthetic distribution shifts, but DETECBENCH is currently limited to synthetic corruptions. Finally, our choice of general-purpose adversarial attacks (BIM, PGD) for object detection, while offering consistency, may overlook task-specific vulnerabilities. Expanding to include dedicated object detection attacks is a key direction for future work.

## REPRODUCIBILITY STATEMENT

We are committed to the complete open-sourcing of our work to ensure reproducibility and community engagement. However, doing so under anonymity (as required) is difficult. For now, we share this Anonymous Repository, which includes the code and evaluations. Most model checkpoints are already available in mmsegmentation and mmdetection. We will additionally share the remaining checkpoints and the generated 3D Common Corruption images upon acceptance.

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

# Benchmarking Reliability and Generalization Beyond Classification

## Paper #18439 Supplementary Material

TABLE OF CONTENT

The supplementary material covers the following information:

**Additional analysis using benchmarking results:**

**Additional Details on the Benchmarking using** SEMSEGBENCH **and Evaluations:**

- Appendix G: In detail explanation of the available functionalities of the SEMSEGBENCH benchmarking tool and description of the arguments for each function.

- Appendix H: **Discussing Adversarially Trained Semantic Segmentation Methods**: We discuss that efforts invested into increasing the reliability of semantic segmentation methods under adversarial attacks can be fruitful as done by some previous works, but there still exists a large gap in reliability that needs to be covered by future work focused in this direction.

- Appendix I: Here we provide additional results from the benchmark evaluated using SEMSEGBENCH.

  - Appendix I.1.1: Evaluations for all models against PGD, SegPGD, and CosPGD attacks under $\ell_\infty$-norm bound and $\ell_2$-norm bound, as a non-targeted attack for the ADE20K, Cityscapes, and PASCAL VOC2012 datasets.

  - Appendix I.2: Evaluations for all models under 2D Common Corruptions at severity 3, for the ADE20K, Cityscapes, and PASCAL VOC2012 datasets.

**Additional Details on the Benchmarking using** DETECBENCH **and Evaluations:**

- Appendix J: Details for the datasets used.
  - Appendix J.1: MS-COCO
  - Appendix J.2: PASCAL VOC

- Appendix K: Additional implementation details for the evaluated benchmark.

- Appendix L: In detail description of the attacks.

- Appendix M: A comprehensive look-up table for all the object detection methods' model weight and datasets pair available in DETECBENCH and used for evaluating the benchmark.

- Appendix N: In detail explanation of the available functionalities of the DETECBENCH benchmarking tool and description of the arguments for each function.

- Appendix O: Here we provide additional results from the benchmark evaluated using DETECBENCH.

  - Appendix O.1: Evaluation using the limited available PASCAL VOC trained models.

  - Appendix O.2: All evaluations using the MS-COCO trained models.

    * Appendix O.2.1: Evaluations for all models against FGSM attack under $\ell_\infty$-norm bound as non-targeted attack.

    * Appendix O.2.2: Evaluations for all models against BIM and PGD attack under $\ell_\infty$-norm bound as non-targeted attack, over multiple attack iterations.

    * Appendix O.2.3: Evaluations for all models under 2D Common Corruptions at severity=3.

    * Appendix O.2.3: Evaluations for all models under 3D Common Corruptions at severity=3.

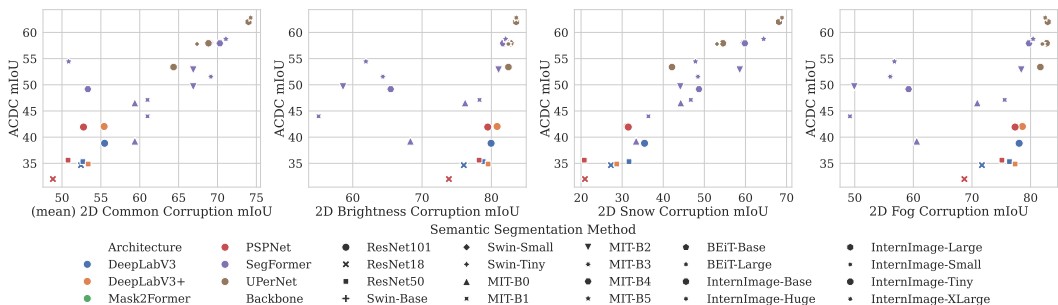

Figure 4: To empirically determine if synthetic common corruptions such that those proposed by Hendrycks & Dietterich (2019) truly represent the distribution and domain shifts in the real world we try to find correlations in evaluations on ACDC and 2D Common Corruptions. Each model is trained on the training dataset of the Cityscapes dataset. The y-axis represents values from evaluations on the ACDC dataset, and the x-axis represents values from evaluations on the Common Corruptions at severity=3. Starting from the left, we find correlations between ACDC the following: first the mean performance across all common corruptions; second the synthetic brightness corruption; third the synthetic snow corruption; and fourth the synthetic fog corruption. We observe a positive correlation, and strong positive correlation between performance on the ACDC and mean performance against all synthetic common corruption.

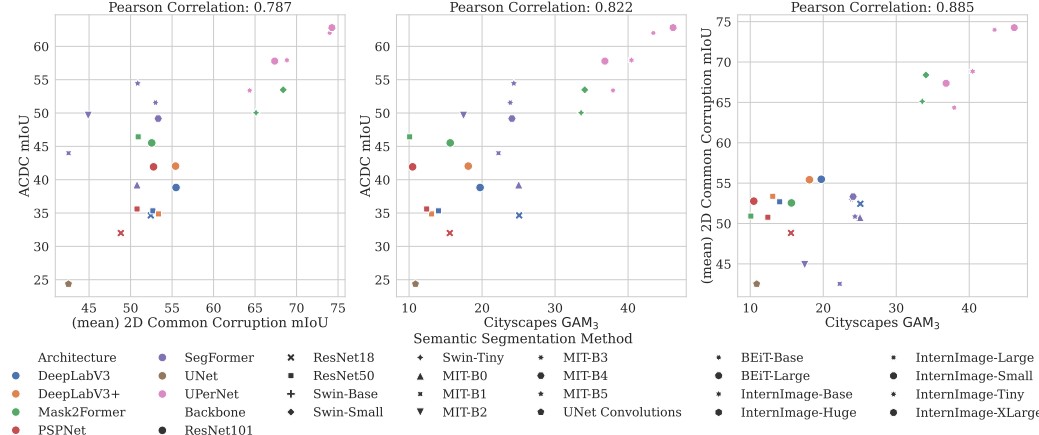

Figure 5: To better understand the correlations from Figure 4, here we look at the correaltions between ACDC mIoU, $GAM_3$, and the mean mIoU across all 2D Common Corruptions.

## A    ADDITIONAL ANALYSIS ON SEMANTIC SEGMENTATION

### A.1    ARE SYNTHETIC CORRUPTIONS USEFUL?

We attempt to study whether synthetic corruptions like those introduced by (Hendrycks & Dietterich, 2019) do represent the distribution shifts in the real world. While this assumption has driven works such as (Hendrycks & Dietterich, 2019; Kar et al., 2022; Kamann & Rother, 2020), to the best of our knowledge, it has not yet been proven. Previous works on robustness (Guo et al., 2023) simply report performance on both; thus, to save compute in the future, we prove this assumption in Fig. 4 and Fig. 5.

For this analysis, we used methods trained on the training set of Cityscapes and evaluated them on 2D Common Corruptions (Hendrycks & Dietterich, 2019) and the ACDC datasets. ACDC is the Adverse Conditions Dataset with Correspondences, consisting of images from regions and scenes similar to Cityscapes but captured under different conditions, such as Day/Night, Fog, Rain, and Snow. These are corruptions in the real world. Thus, we attempt to find correlations between performance against synthetic corruptions from 2D Common Corruptions (severity=3) and ACDC. We

analyze each common corruption separately and also the mean performance across all 2D Common Corruptions.

In Fig. 4 and Fig. 5, we observe a very strong positive correlation in performance against ACDC and mean performance across all 2D Common Corruptions. This novel finding helps the community significantly, as this means that we do not need to go into the wild to capture images with distribution shifts, as synthetic corruptions serve as a reliable proxy for real-world conditions. Since some synthetic corruptions attempt to directly mimic the real-world scenarios in ACDC, like changes in lighting due to Day/Night changes or changes in weather due to snowfall or fog, we analyze the correlation of relevant corruptions to ACDC. We find that there exists a weak positive correlation between performance against ACDC and performance against Brightness corruption and Fog corruption. Interestingly, there is a strong positive correlation in performance against ACDC and Snow Corruption. These positive correlations further strengthen the argument of using synthetic common corruptions over investing effort to capture these corruptions in the wild.

## A.2 CORRELATION ACROSS DATASETS

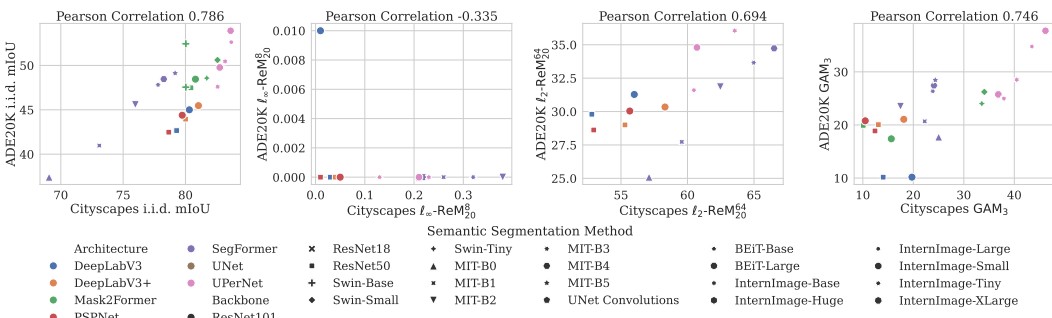

Figure 6: To find correlations in observations across datasets, here we use the ADE20K dataset and Cityscapes dataset. Each model is trained on the training dataset of the respective dataset on which it is eventually evaluation. The y-axis represents values from evaluations on the ADE20K dataset, and the x-axis represents values from evaluations on the Cityscapes dataset. Starting from the left, we find correlations between ADE20K and Cityscapes dataset for have the following: first the i.i.d. performance; second the reliability under $\ell_\infty$-norm bounded adversarial attacks; third the reliability under $\ell_2$-norm bounded adversarial attacks; and fourth the generalization ability.

For ease of understanding, the analysis thus far was limited to the recent and commonly used ADE20K dataset (Zhou et al., 2019). However, there exist other commonly used datasets such as Cityscapes (Cordts et al., 2016) and PASCAL VOC2012 (Everingham et al., 2012). Each dataset brings in a different challenge with it, such as different scenes or different numbers of classes. We attempt to find if a correlation exists between the model's performance, reliability, and generalization abilities when using different datasets. In Fig. 6, we compare models trained on the training dataset on the respective datasets used for evaluations. We observe that there appears to be a high correlation in the i.i.d. performance of models between the two datasets with no outliers. While the value of the metrics is slightly lower for ADE20K compared to Cityscapes, a weak correlation also exists for reliability under $\ell_2$-norm adversarial attacks. Under $\ell_\infty$-norm adversarial attacks, the values are very close to zero, impeding a meaningful correlation study. For the generalization ability, there appears to be a weak positive correlation between the OOD Robustness evaluations using ADE20K and Cityscapes. Please refer to the appendix for results with PASCAL VOC2012.

This study helps us understand that given a correlation in performance across datasets, future works need not invest compute resources in exhaustive studies across datasets, especially for i.i.d. performance.

## A.3 CORRELATION IN METRICS

Here, we show a high positive correlation in the different metrics captured for all the three considered datasets: ADE20K, Cityscapes, and PASCAL VOC2012, justifying only using mIoU for all the

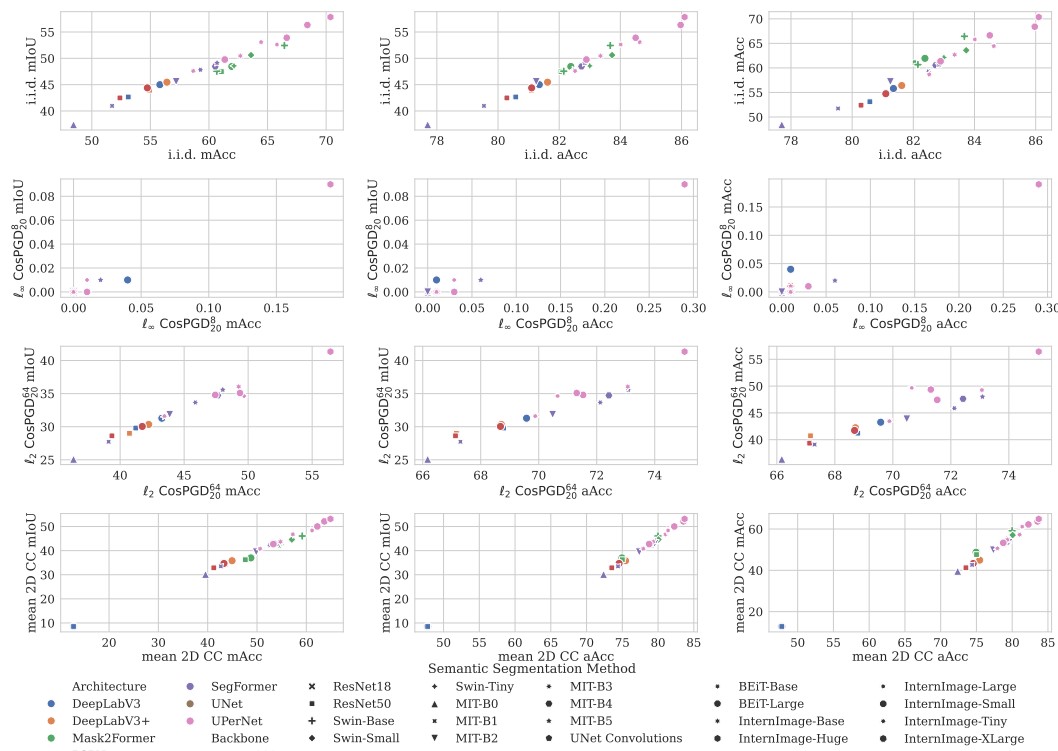

Figure 7: **Dataset used: ADE20K**. The performance of semantic segmentation methods is usually captured across different metrics, namely: mIoU, mAcc, and aAcc. In the analysis of this work, we only used mIoU. Here we show that there is a high positive correlation between these metrics, and observations made using mIoU would still hold using other metrics. The top row is correlation when using i.i.d. data for evaluations. The second row is when using $\ell_\infty$-norm bounded CosPGD attack with $\epsilon = \frac{8}{255}$. The third row is when using $\ell_2$-norm bounded CosPGD attack with $\epsilon = \frac{8}{255}$. The fourth row is when using 2D Common Corruptions, here we calculate the mean for each metric across all 2D Common Corruptions. Colors are used to show different architectures and marker styles are used to show different backbones used by the semantic segmentation methods.

analysis. We show this for ADE20K in Figure 7, Cityscapes in Figure 8, and PASCAL VOC2012 in Figure 9. We observe a very strong positive correlation between the different metrics: mIoU, mAcc, and aAcc. Thus, the analysis made using mIoU would also hold if made using other metrics.

### A.3.1 EXTENSION TO PERFORMANCE V/S RELIABILITY V/S GENERALIZATION

As discussed in Sec. 1, semantic segmentation methods proposed over the years have improved their performance on i.i.d. samples. However, due to limited works towards robustness of semantic segmentation methods, we lack a large-scale analysis for correlations between their performance and reliability, performance and generalization ability, and reliability and generalization abilities. To gather this crucial information, we analyze these important correlations in Fig. 2a. Here, we first reiterate the observations made by (Gu et al., 2022; Agnihotri et al., 2024b) that semantic segmentation methods are not inherently robust to strong adversarial attacks, especially those bounded by the $\ell_\infty$-norm.

Second, we also observe that typical $\ell_\infty$-norm attacks with $\epsilon = \frac{8}{255}$ as used by (Agnihotri et al., 2024b; Gu et al., 2022) are simply too strong to analyze correlations. They cause almost all semantic segmentation methods to completely fail. Thus, we observe that when using $\ell_\infty$-norm attacks to measure reliability, there exists no correlation between reliability and other aspects. This raises credible concerns regarding the reliability of semantic segmentation methods under $\ell_\infty$-norm attacks. Some methods like (Xu et al., 2021; Croce et al., 2024) attempt to address this. However, semantic segmentation methods seem slightly more robust to $\ell_2$-norm bounded adversarial attacks.

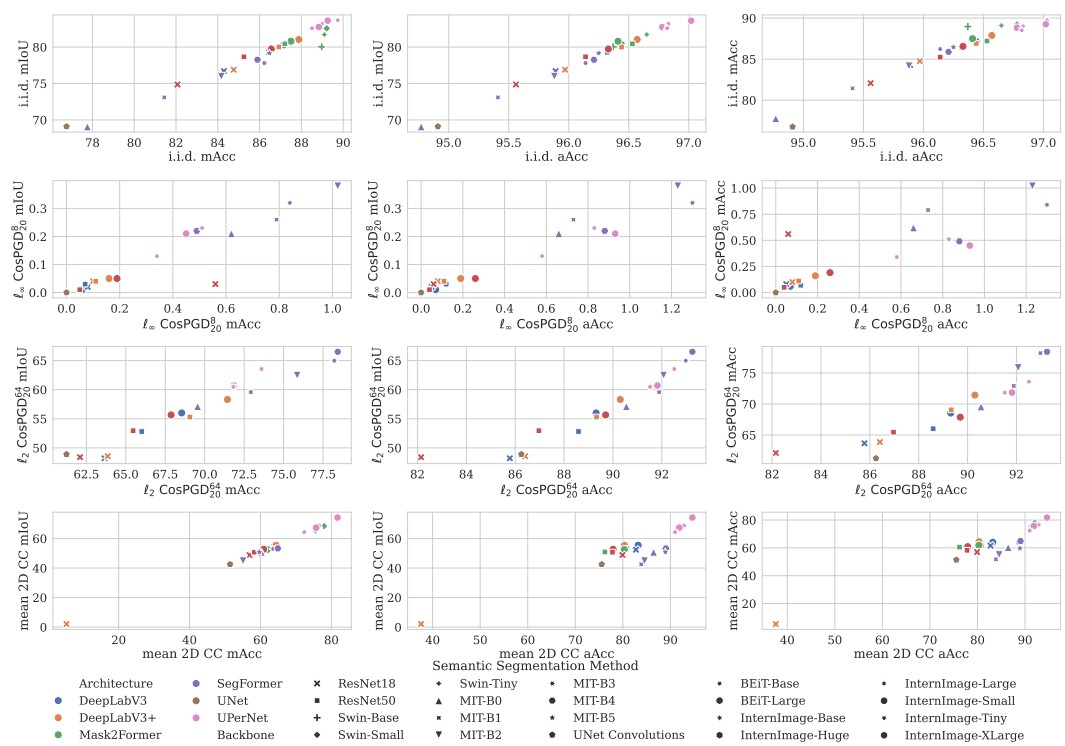

Figure 8: **Dataset used: Cityscapes**. The performance of semantic segmentation methods is usually captured across different metrics, namely: mIoU, mAcc, and aAcc. In the analysis of this work, we only used mIoU. Here we show that there is a high positive correlation between these metrics, and observations made using mIoU would still hold using other metrics. The top row is correlation when using i.i.d. data for evaluations. The second row is when using $\ell_\infty$-norm bounded CosPGD attack with $\epsilon = \frac{8}{255}$ and $\alpha$=0.01. The third row is when using $\ell_2$-norm bounded CosPGD attack with $\epsilon = 64$ and $\alpha$=0.1. The fourth row is when using 2D Common Corruptions, here we calculate the mean for each metric across all 2D Common Corruptions. Colors are used to show different architectures and marker styles are used to show different backbones used by the semantic segmentation methods.

For reliability under $\ell_2$-norm attacks, we find that there does exist a strong correlation between i.i.d. performance and reliability.

Next, we find some correlation between i.i.d. performance and generalization ability. Some methods like InternImage achieve both descent i.i.d. performance and OOD robustness. However, methods such as DeepLabV3 (Chen et al., 2017) do not appear to have a good generalization ability. Please note, that even InternImage, while generalizing better than the other models, still lacks true generalization abilities, which would ensure no drop in performance between i.i.d. and OOD samples. However, we see a significant drop for InternImage. Additionally, we observe that while SegFormer (Xie et al., 2021) and InternImage (Wang et al., 2023b) are not robust against $\ell_\infty$-norm attacks, they are robust to some extent against $\ell_2$-norm attack when using large backbones (relatively higher number of parameters). This is in line with the observations made for image classification models, where models with a large number of parameters were found to be more robust (Croce et al., 2021) and will be subject to discussion in the following subsection.

For reliability and generalization ability, we observe a lack of high correlation between the two, even when using $\ell_2$-norm as the reliability measure. InternImage models are better against Common Corruptions. **Thus, future methods addressing one aspect, reliability or OOD generalization, might not inherently address the other, and each aspect requires intentional focus.**

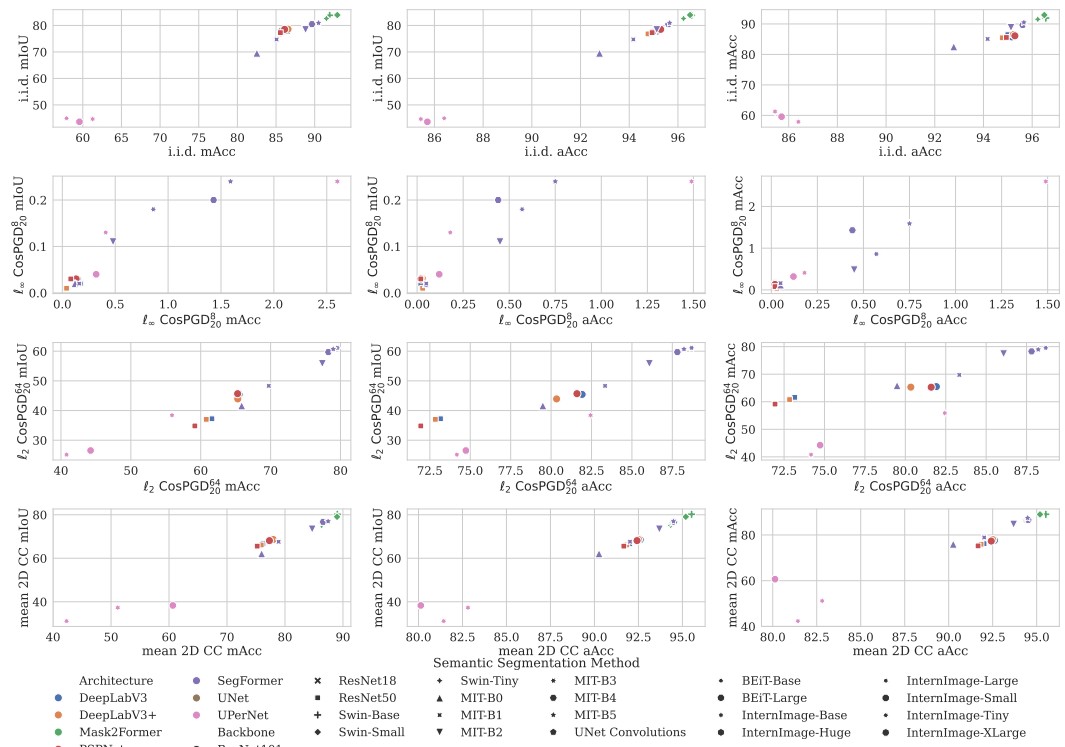

Figure 9: **Dataset used: PASCAL VOC2012**. The performance of semantic segmentation methods is usually captured across different metrics, namely: mIoU, mAcc, and aAcc. In the analysis of this work, we only used mIoU. Here we show that there is a high positive correlation between these metrics, and observations made using mIoU would still hold using other metrics. The top row is correlation when using i.i.d. data for evaluations. The second row is when using $\ell_\infty$-norm bounded CosPGD attack with $\epsilon = \frac{8}{255}$. The third row is when using $\ell_2$-norm bounded CosPGD attack with $\epsilon = \frac{8}{255}$. The fourth row is when using 2D Common Corruptions, here we calculate the mean for each metric across all 2D Common Corruptions. Colors are used to show different architectures and marker styles are used to show different backbones used by the semantic segmentation methods.

# B    ADDITIONAL ANALYSIS ON OBJECT DETECTION

## B.1    PROPOSAL PREDICTION METHOD

Traditional object detection methods such as Faster-RCNN (Ren, 2015), and RetinaNet (Lin, 2017), after feature extraction, used region-based proposals that required Non-Maximum Suppression for detecting objects in a scene, and due to these two-stages and use of an anchor bounding box, these methods were classified as 'Anchor-based Two Stage' object detection methods. These methods were followed by 'Anchor-based One stage' methods that made predictions directly using the features extracted and then 'Anchor-free One stage' methods. These can be further categorized based on the specific technique used for detecting objects, for example, ATSS (Zhang et al., 2020b) uses a Center-based approach, while CenterNet (Duan et al., 2019) uses a keypoint triplet and thus is a keypoint-based method, and TOOD (Feng et al., 2021) presents a unique task-aligned way for detecting objects, and RTMDet (Lyu et al., 2022) is a unique one-stage method that improves upon YOLO series (Redmon & Farhadi, 2017; Ge, 2021; Li et al., 2022a; Wang et al., 2023a; Jocher et al., 2020; Redmon et al., 2016; Bochkovskiy et al., 2020; Redmon & Farhadi, 2018).

Recently, DETR (Carion et al., 2020) proposed a unique attention-mechanism-based system that replaces anchors with object queries. This inspired many works such as Co-DETR (Zong et al., 2023), DINO (Caron et al., 2021), DDQ-DETR (Zhang et al., 2023), and others that essentially use 'Attention-based Object Queries' for object detection with new variants and constraints.

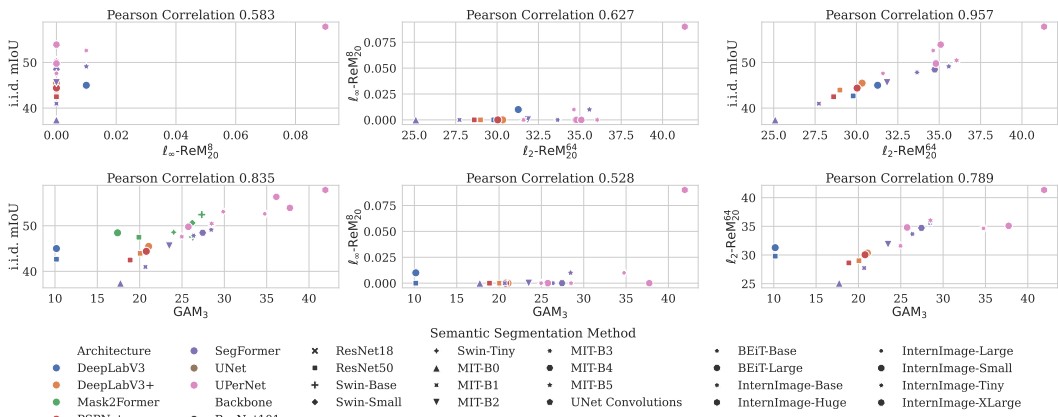

Figure 10: **Semantic Segmentation:** Using the ADE20K dataset, here we analyze correlations in i.i.d. performance, reliability and generalization abilities of different methods. The colors represent the architecture of the method, while the shapes of the markers represent the backbone of the respective method. This figure shows the correlations between the following: i.i.d. performance and reliability under $\ell_\infty$-norm bounded adversarial attacks in top row left; reliability under $\ell_\infty$-norm bounded adversarial attacks and reliability under $\ell_2$-norm bounded adversarial attacks in top row middle; i.i.d. performance and reliability under $\ell_2$-norm bounded adversarial attacks in top row right; i.i.d. performance and generalization ability in bottom row left; reliability under $\ell_\infty$-norm bounded adversarial attacks and generalization ability in bottom row center; and reliability under $\ell_2$-norm bounded adversarial attacks and generalization ability in bottom row right. All methods were trained on the train set of the ADE20K dataset.

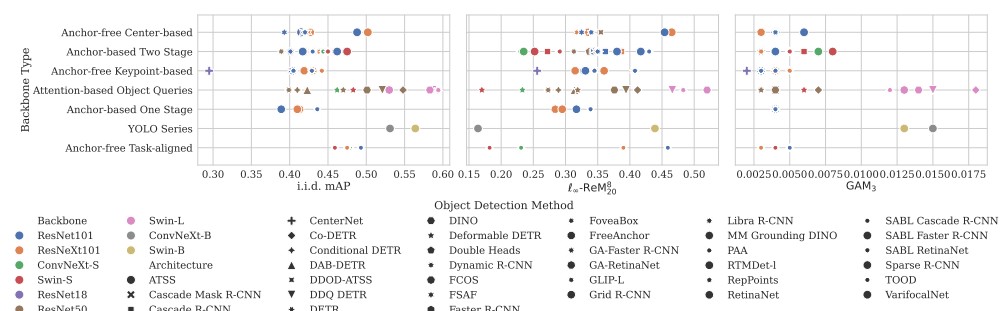

Figure 11: Using the MS-COCO dataset, here we analyze i.i.d. performance, reliability, and generalization abilities based on the type of proposal technique used by the object detection methods. The colors represent the backbone of the respective method, while different marker shapes represent the architecture of the method. On the y-axis, all the proposal techniques as listed as 'Proposal Type'. On the x-axis, starting from the left have the following: first the i.i.d. performance; second the reliability under $\ell_\infty$-norm bounded adversarial attacks; third the generalization ability. All methods were trained on the train set of the MS-COCO dataset.

In Fig. 11, we attempt to study these important design choices made in an object detection method and their impact on i.i.d. performance, reliability under adversarial attacks, and generalization ability to OOD samples.

We observe that the recently proposed methods that use 'Attention-based Object Queries' have the highest i.i.d. performance. However, barring the few instances of 'Attention-based Object Queries' that use a large Swin backbone, these methods do not outperform other proposal prediction methods in terms of reliability and generalization abilities. Interestingly, the YOLO series method, TOOD (Feng et al., 2021) has a rather high generalization ability, but its reliability depends on the backbone. With a Swin-B backbone, the reliability of TOOD is relatively high, but with a ConvNeXt-B backbone, the reliability is among the lowest.

## B.2 CORRELATION BETWEEN 2D AND 3D CORRUPTIONS

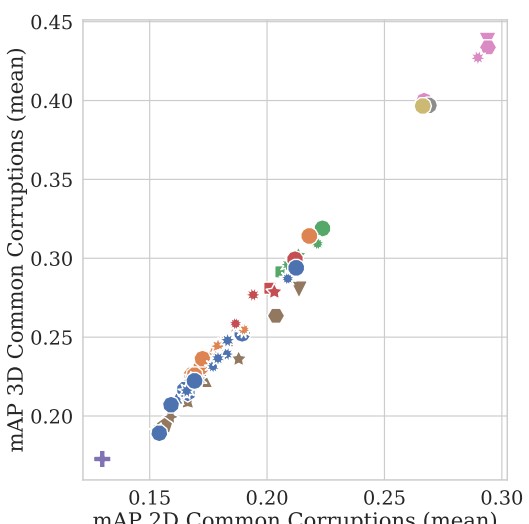

Figure 12: Applying 3D Common Corruptions (y-axis) and 2D Common Corruptions (x-axis) to the MS-COCO dataset and finding a strong positive correlation in the performance of all object detection methods measured by the mean mAP values across all respective corruptions. Here the symbolic representations are the same as Fig. 3a, Fig. 3b, Fig. 11, and Fig. 3c.

Computing the generalization ability of object detection methods against common corruptions (Hendrycks & Dietterich, 2019; Kar et al., 2022) can be computationally expensive, as we evaluate against 15 2D Common Corruptions and 10 3D Common Corruptions. However, if there exists a very strong correlation between their performance, then future object detection methods, when reporting relative generalization abilities, can avoid computing on both 2D and 3D Common Corruptions, especially on 3D Common Corruptions, as these are more expensive to compute. This is because 3D Common Corruptions take depth information into account and attempt to simulate lighting conditions and behavior of a 3D environment. While more realistic, these renderings are time and compute-wise very expensive. We observe in Fig. 12 that, indeed, there is a very strong correlation between the mean mAP over all 3D Common Corruptions and the mean mAP over all 2D Common Corruptions. Thus, unless specifically addressing the lack of generalization of object detection methods to specific 3D Common Corruptions such as xy motion blur, z motion blur, or fog corruption, future object detection methods can focus on computing merely robustness to 2D Common Corruptions to analyze generalization abilities relative to other object detection methods.

## B.3 CORRELATION IN METRICS

Here, we show a high positive correlation in the different metrics captured for the MS-COCO dataset, justifying only using mAP for all the analysis. We show this in Figure 13 and observe a very strong positive correlation between different metrics.

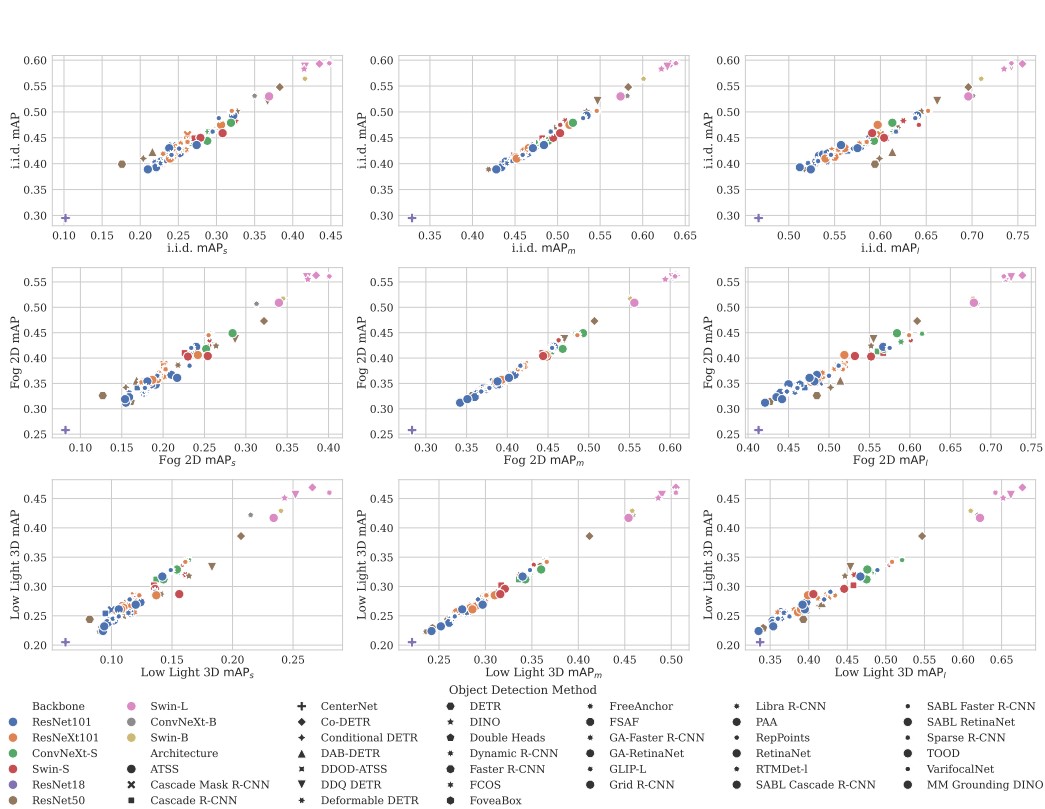

Figure 13: MS-COCO dataset is usually captured across different metrics, namely: mAP, $mAP_s$, $mAP_m$, and $mAP_l$. In the analysis of this work, we only used mAP. Here we show that there is a high positive correlation between these metrics, and observations made using mAP would still hold using other metrics. The top row is correlation when using i.i.d. data for evaluations. The middle row is when using a randomly chosen 2D Common Corruption, Fog 2D. The bottom row is when using a randomly chosen 3D Common Corruption Low Light. Colors are used to show different backbones and marker styles are used to show different architectures used by the object detection methods.

# Additional Details on the Benchmarking using SEMSEGBENCH and Evaluations (All Benchmarking Results):

## TABLE OF CONTENT

The supplementary material covers the following information:

## C    DATASET DETAILS

SEMSEGBENCH supports a total of three distinct semantic segmentation datasets. Following, we describe these datasets in detail.

### C.1 ADE20K

ADE20K (Zhou et al., 2019) dataset contains pixel-level annotations for 150 object classes, with a total of 20,210 images for training, 2000 images for validation, and 3000 images for testing. Following common practice (Agnihotri et al., 2024b; Xie et al., 2021) we evaluate using the validation images.

### C.2 CITYSCAPES

The Cityscapes dataset (Cordts et al., 2016) comprises a total of 5000 images sourced from 50 different cities in Germany and neighboring countries. The images were captured at different times of the year and under typical meteorological conditions. Each image was subject to pixel-wise annotations by human experts. The dataset is split into three subsets: training (2975 images), validation (500 images), and testing (1525 images). This dataset has pixel-level annotations for 30 object classes.

### C.3 PASCAL VOC2012

The PASCAL VOC 2012 (Everingham et al., 2012), contains 20 object classes and one background class, with 1464 training images, and 1449 validation images. We follow common practice (Hariharan et al., 2015; Gu et al., 2022; Zhao, 2019; Zhao et al., 2017), and use work by (Hariharan et al., 2011), augmenting the training set to 10,582 images. We evaluate using the validation set.

## D    IMPLEMENTATION DETAILS OF THE BENCHMARK

Following we provide details regarding the experiments done for creating the benchmark used in the analysis.

**Compute Resources.** Most experiments were done on a single 40 GB NVIDIA Tesla V100 GPU each, however, SegFormer (Xie et al., 2021) and Mask2Former (Cheng et al., 2022) with large backbones are more compute-intensive, and thus 80GB NVIDIA A100 GPUs or NVIDIA H100 were used for these models, a single GPU for each experiment. Training some of the architectures with large backbones required using two to four GPUs in parallel.

**Datasets Used.** Performing adversarial attacks and OOD robustness evaluations are very expensive and compute-intensive. Thus, for the benchmark, we only use ADE20k, Cityscapes, and PASCAL VOC2012 as these are the most commonly used datasets for evaluation (Agnihotri et al., 2024b; Xie et al., 2021; Cheng et al., 2022; Zhao et al., 2017; Kamann & Rother, 2020).

**Metrics Calculation.** In Sec. 3 we introduce two new metrics for better understanding our analysis, given the large scale of the benchmark created. For calculating $\mathrm{ReM}$ values we used PGD, SegPGD, and CosPGD attack with step size $\alpha$=0.01, perturbation budget $\epsilon = \frac{8}{255}$ under the $\ell_\infty$-norm bound, as non-targeted attacks. Under the $\ell_2$-norm bound, we use $\epsilon$=64, and $\alpha$=0.1, as also used by (Agnihotri et al., 2024b). We use 20 attack iterations for calculating the $\mathrm{ReM}$ values because as shown by (Agnihotri et al., 2024b) and (Schmalfuss et al., 2022b), 20 iterations are enough to optimize an attack to truly understand the performance of the attacked method. For calculating $\mathrm{GAM}$, we use all 15 2D Common Corruptions: 'Gaussian Noise', Shot Noise', 'Impulse Noise', 'Defocus Blur', 'Frosted Glass Blur', 'Motion Blur', 'Zoom Blur', 'Snow', 'Frost', 'Fog', 'Brightness', 'Contrast', 'Elastic Transform', 'Pixelate', 'JPEG Compression'. All the common corruptions are at severity 3.

**Calculating the mIoU.** mIoU is the mean Intersection over Union of the predicted segmentation mask with the ground truth segmentation mask.

**Other Metrics.** Apart from mIoU, SEMSEGBENCH also enables calculating the mean accuracy over all pixels (mAcc) and the mean accuracy over all classes (allAcc).

**Models Used.** All available checkpoints, as shown in Tab. 2 for ADE20K, Cityscapes, and PAS-CAL VOC2012 were used for creating the benchmark, these methods include some of the first efforts in DL-based semantic segmentation methods like UNet (Ronneberger et al., 2015), and some of the most recent SotA methods like InterImage (Wang et al., 2023b).

# E  DESCRIPTION OF SEMSEGBENCH

Following, we describe the benchmarking tool, SEMSEGBENCH. It is built using mmsegmentation (Contributors, 2020), and all architectures, backbones, and datasets supported by mmsegmentation (please refer Appendix C for additional details on the datasets). It enables training and evaluations on all aforementioned combinations including evaluations using SotA adversarial attacks such as CosPGD (Agnihotri et al., 2024b) and SegPGD (Gu et al., 2022), and other commonly used adversarial attacks like FGSM (Goodfellow et al., 2015), and PGD (Kurakin et al., 2017) under various lipshitz ($l_p$) norm bounds.

Additionally, it enables evaluations for Out-of-Distribution (OOD) robustness by corrupting the inference samples using 2D Common Corruptions (Hendrycks & Dietterich, 2019).

We follow the nomenclature set by RobustBench (Croce et al., 2021) and use "threat_model" to define the kind of evaluation to be performed. When "threat_model" is defined to be "None", the evaluation is performed on unperturbed and unaltered images, if the "threat_model" is defined to be an adversarial attack, for example "PGD", "CosPGD" or "SegPGD", then SEMSEGBENCH performs an adversarial attack using the user-defined parameters. We elaborate on this in Appendix E.1. Whereas, if "threat_model" is defined to be "2DCommonCorruptions", the SEMSEGBENCH performs evaluations after perturbing the images with 2D Common Corruptions. We elaborate on this in Appendix E.2.

If the queried evaluation already exists in the benchmark provided by this work, then SEMSEG-BENCH simply retrieves the evaluations, thus saving computation.

## E.1  ADVERSARIAL ATTACKS

Due to significant similarity, most of the text here has been adapted from (Agnihotri et al., 2025). SEMSEGBENCH enables the use of all the attacks mentioned in Sec. 2 to help users better study the reliability of their semantic segmentation methods. We choose to specifically include these white-box adversarial attacks as they either serve as the common benchmark for adversarial attacks in classification literature (FGSM, PGD) or they are unique attacks proposed specifically for pixel-wise prediction tasks (CosPGD) and semantic segmentation (SegPGD). These attacks are currently designed to be *Non-targeted* which simply fool the model into making incorrect predictions, irrespective of what the model eventually predicts. Attacks can also be *Targeted*, where the model is fooled to make a certain prediction, we intend to add this functionality in future iterations of SEM-SEGBENCH.

Following, we discuss these attacks in detail and highlight their key differences.

**FGSM.** Assuming a non-targeted attack, given a model $f_\theta$ and an unperturbed input sample $\boldsymbol{X}^{\text{clean}}$ and ground truth label $\boldsymbol{Y}$, FGSM attack adds noise $\delta$ to $\boldsymbol{X}^{\text{clean}}$ as follows,

$$\boldsymbol{X}^{\text{adv}} = \boldsymbol{X}^{\text{clean}} + \alpha \cdot \text{sign}\nabla_{\boldsymbol{X}^{\text{clean}}} L(f_\theta(\boldsymbol{X}^{\text{clean}}), \boldsymbol{Y}), \tag{1}$$

$$\delta = \phi^\epsilon(\boldsymbol{X}^{\text{adv}} - \boldsymbol{X}^{\text{clean}}), \tag{2}$$

$$\boldsymbol{X}^{\text{adv}} = \phi^r(\boldsymbol{X}^{\text{clean}} + \delta). \tag{3}$$

Here, $L(\cdot)$ is the loss function (differentiable at least once) which calculates the loss between the model prediction and ground truth, $\boldsymbol{Y}$. $\alpha$ is a small value of $\epsilon$ that decides the size of the step to be taken in the direction of the gradient of the loss w.r.t. the input image, which leads to the input

sample being perturbed such that the loss increases. $\boldsymbol{X}^{\mathrm{adv}}$ is the adversarial sample obtained after perturbing $\boldsymbol{X}^{\mathrm{clean}}$. To make sure that the perturbed sample is semantically indistinguishable from the unperturbed clean sample to the human eye, steps from Eq. (2) and Eq. (3) are performed. Here, function $\phi^\epsilon$ is clipping the $\delta$ in $\epsilon$-ball for $\ell_\infty$-norm bounded attacks or the $\epsilon$-projection in other $l_p$-norm bounded attacks, complying with the $\ell_\infty$-norm or other $l_p$-norm constraints, respectively. While function $\phi^r$ clips the perturbed sample ensuring that it is still within the valid input space. FGSM, as proposed, is a single step attack. For targeted attacks, $\boldsymbol{Y}$ is the target and $\alpha$ is multiplied by -1 so that a step is taken to minimize the loss between the model's prediction and the target prediction, we intend to add this option in future iterations of SEMSEGBENCH.

**BIM.**  This is the direct extension of FGSM into an iterative attack method. In FGSM, $\boldsymbol{X}^{\mathrm{clean}}$ was perturbed just once. While in BIM, $\boldsymbol{X}^{\mathrm{clean}}$ is perturbed iteratively for time steps $t \in [0, \boldsymbol{T}]$, such that $t \in \mathbb{Z}^+$, where $\boldsymbol{T}$ are the total number of permissible attack iterations. This changes the steps of the attack from FGSM to the following,

$$\boldsymbol{X}^{\mathrm{adv}_{t+1}} = \boldsymbol{X}^{\mathrm{adv}_t} + \alpha \cdot \mathrm{sign}\nabla_{\boldsymbol{X}^{\mathrm{adv}_t}} L(f_\theta(\boldsymbol{X}^{\mathrm{adv}_t}), \boldsymbol{Y}), \tag{4}$$

$$\delta = \phi^\epsilon(\boldsymbol{X}^{\mathrm{adv}_{t+1}} - \boldsymbol{X}^{\mathrm{clean}}), \tag{5}$$

$$\boldsymbol{X}^{\mathrm{adv}_{t+1}} = \phi^r(\boldsymbol{X}^{\mathrm{clean}} + \delta). \tag{6}$$

Here, at $t{=}0$, $\boldsymbol{X}^{\mathrm{adv}_t}{=}\boldsymbol{X}^{\mathrm{clean}}$.

**PGD.**  Since in BIM, the initial prediction always started from $\boldsymbol{X}^{\mathrm{clean}}$, the attack required a significant amount of steps to optimize the adversarial noise and yet it was not guaranteed that in the permissible $\epsilon$-bound, $\boldsymbol{X}^{\mathrm{adv}_{t+1}}$ was far from $\boldsymbol{X}^{\mathrm{clean}}$. Thus, PGD proposed introducing stochasticity to ensure random starting points for attack optimization. They achieved this by perturbing $\boldsymbol{X}^{\mathrm{clean}}$ with $\mathcal{U}(-\epsilon, \epsilon)$, a uniform distribution in $[-\epsilon, \epsilon]$, before making the first prediction, such that, at $t{=}0$

$$\boldsymbol{X}^{adv_t} = \phi^r(\boldsymbol{X}^{clean} + \mathcal{U}(-\epsilon, \epsilon)). \tag{7}$$

**APGD.**  Auto-PGD (Wong et al., 2020) is an effective extension to the PGD attack that effectively scales the step size $\alpha$ over attack iterations considering the compute budget and the success rate of the attack.

**SegPGD.**  SegPGD (Gu et al., 2022) is an effective white-box adversarial attack proposed specifically for semantic segmentation methods. It optimizes the PGD attack by splitting the pixel-wise predictions into correctly classified and wrongly classified and then scales the loss differently for these two categories of pixels over attack optimization iterations (steps). The intent of this separation and different scaling is that in initial attack optimization iterations there would be many correctly classified pixels and only a few incorrectly classified pixels, thus if the loss for the correctly classified pixels is scaled higher and the loss of the incorrectly classified pixels is scaled lower then the attack focuses on changing model predictions on pixels that are correctly classified rather than focusing on increasing the loss on already incorrectly classified pixels. As the attack iterations increase, reaching their limit, ideally, there exist more incorrectly classified pixels than correctly classified ones, in this case, the loss of the incorrectly classified pixels is scaled higher, while the loss for the correctly classified pixels is scaled lower so that the attack can still optimize further. The optimization strategy of SegPGD can be summarized using Eq. (8).

$$\boldsymbol{X}^{\mathrm{adv}_{t+1}} = \boldsymbol{X}^{\mathrm{adv}_t} + \alpha \cdot \mathrm{sign}\nabla_{\boldsymbol{X}^{\mathrm{adv}_t}} \left( \sum_i \left( 1 - \left| \lambda - \frac{|(argmax(f_\theta(\boldsymbol{X}^{\mathrm{adv}_t})_i) - \Psi(\boldsymbol{Y}_i)|}{2} \right| \right) \right.$$
$$\left. \cdot \mathcal{L}\left(f_\theta(\boldsymbol{X}^{\mathrm{adv}_t})_i, \boldsymbol{Y}_i\right) \right) \tag{8}$$

for all locations $i \in P^T \cup P^F$, i.e. $|\lambda - |(argmax(f(\boldsymbol{X}^{\mathrm{adv}_t})) - \boldsymbol{Y}|/2|$ equals $1 - \lambda$ for incorrect predictions, it equals $\lambda$ for correct predictions, and $\Psi(\cdot)$ is used to one-hot encode the labels. Here $P^T$ are the correctly classified pixels and $P^F$ are the incorrectly classified pixels, and $\lambda$ is a scaling factor set heuristically. $\mathcal{L}$ is explained in Eq. (9).

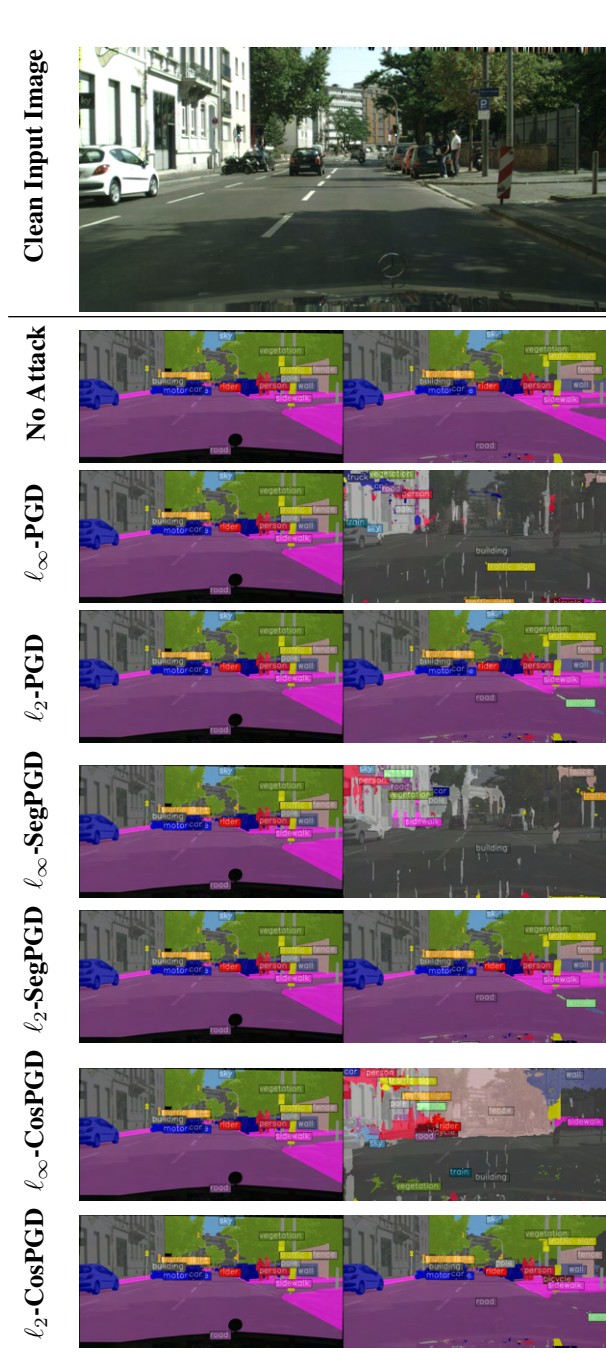

Figure 14: Illustrating changes in prediction due to different $\ell_\infty$-norm and $\ell_2$-norm bounded attacks on a randomly chosen input image from the **Cityscapes dataset**, when attaching the semantic segmentation method **InterImage-Base**. In the subfigures with semantic segmentation mask predictions, **Left: Ground Truth Mask**, and **Right: Predicted Mask**.

**CosPGD.** Almost all previously discussed attacks were proposed for the image classification task, or like SegPGD were constrained to one pixel-wise prediction task of semantic segmentation. Here, the input sample is a 2D image of resolution $H \times W$, where H and W are the height and width of the spatial resolution of the sample, respectively. Pixel-wise information is inconsequential for image classification. This led to the pixel-wise loss $\mathcal{L}(\cdot)$ being aggregated to $L(\cdot)$, as follows,

$$L(f_\theta(\boldsymbol{X}^{\text{adv}_t}), \boldsymbol{Y}) = \frac{1}{H \times W} \sum_{i \in H \times W} \mathcal{L}(f_\theta(\boldsymbol{X}^{\text{adv}_t})_i, \boldsymbol{Y}_i). \tag{9}$$

This aggregation of $\mathcal{L}(\cdot)$ fails to account for pixel-wise information available in tasks other than image classification, such as pixel-wise prediction tasks like Optical Flow estimation, Image Restoration, and others. Thus, in their work (Agnihotri et al., 2024b) propose an effective extension of the PGD attack that takes pixel-wise information into account by scaling $\mathcal{L}(\cdot)$ by the alignment between the distribution of the predictions and the distributions of $\boldsymbol{Y}$ before aggregating leading to a better-optimized attack, modifying Eq. (4) as follows,

$$\boldsymbol{X}^{\text{adv}_{t+1}} = \boldsymbol{X}^{\text{adv}_t} + \alpha \cdot \text{sign} \nabla_{\boldsymbol{X}^{\text{adv}_t}} \sum_{i \in H \times W} \cos\left(\psi(f_\theta(\boldsymbol{X}^{\text{adv}_t})_i), \Psi(\boldsymbol{Y}_i)\right) \cdot \mathcal{L}\left(f_\theta(\boldsymbol{X}^{\text{adv}_t})_i, \boldsymbol{Y}_i\right). \tag{10}$$

Where, functions $\psi$ and $\Psi$ are used to obtain the distribution over the predictions and $\boldsymbol{Y}_i$, respectively, and the function $\cos$ calculates the cosine similarity between the two distributions. CosPGD is the unified SotA adversarial attack for pixel-wise prediction tasks.

Fig. 14, shows adversarial examples created using the SotA attacks and how they affect the model predictions.

### E.2 OUT-OF-DISTRIBUTION ROBUSTNESS

Due to significant similarity, most of the text here has been adapted from (Agnihotri et al., 2025). While adversarial attacks help explore vulnerabilities of inefficient feature representations learned by a model, another important aspect of reliability is generalization ability. Especially, generalization to previously unseen samples or samples from significantly shifted distributions compared to the distribution of the samples seen while learning model parameters. As one cannot cover all possible scenarios during model training, a certain degree of generalization ability is expected from models. However, multiple works (Hendrycks & Dietterich, 2019; Kamann & Rother, 2020; Hoffmann et al., 2021) showed that models are surprisingly less robust to distribution shifts, even those that can be caused by commonly occurring phenomena such as weather changes, lighting changes, etc. This makes the study of Out-of-Distribution (OOD) robustness an interesting avenue for research. Thus, to facilitate the study of robustness to such commonly occurring corruptions, SEMSEGBENCH enables evaluation against prominent image corruption methods. Following we describe these methods in detail.

**2D Common Corruptions.** (Hendrycks & Dietterich, 2019) propose introducing a distribution shift in the input samples by perturbing images with a total of 15 synthetic corruptions that could occur in the real world. These corruptions include weather phenomena such as fog, and frost, digital corruptions such as jpeg compression, pixelation, and different kinds of blurs like motion and zoom blur, and noise corruptions such as Gaussian and shot noise, amongst others corruption types. Each of these corruptions can perturb the image at 5 different severity levels between 1 and 5. The final performance of the model is the mean of the model's performance on all the corruptions, such that every corruption is used to perturb each image in the evaluation dataset. Since these corruptions are applied to a 2D image, they are collectively termed 2D Common Corruptions.

We show examples of perturbed images over some corruptions and the changed predictions in Figure 15.

## F MODEL ZOO

The trained checkpoints for all models available in SEMSEGBENCH can be obtained using the following lines of code:

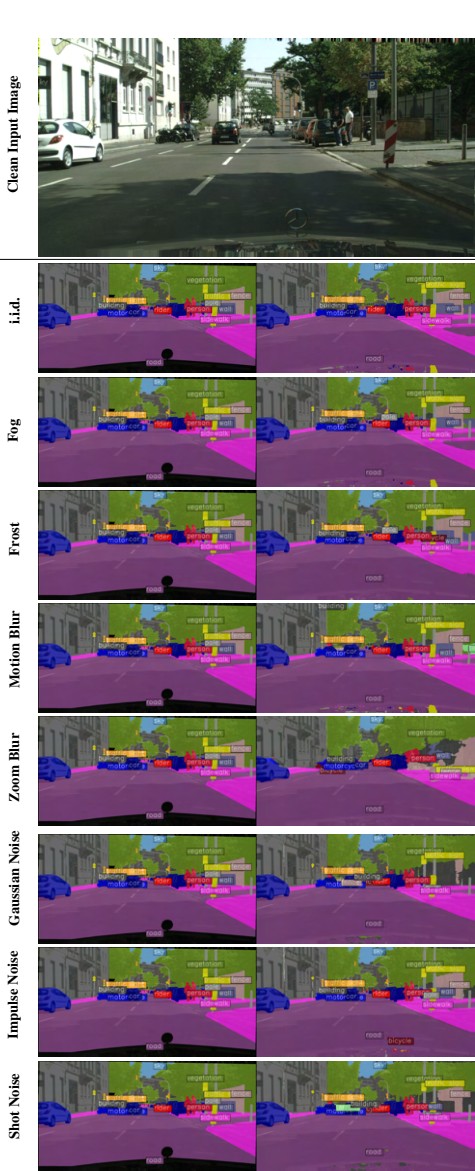

Figure 15: Illustrating changes in prediction due to different 2D Common Corruptions on a randomly chosen input image from the **Cityscapes dataset**, when attaching the semantic segmentation method **InterImage-Base**. In the subfigures with semantic segmentation mask predictions, **Left: Ground Truth Mask**, and **Right: Predicted Mask**.

```
from semsegbench.evals import load_model
model = load_model(model_name='upernet',
  backbone='InterImage-H', dataset='ADE20k')
```

Users need to specify the triplet of architecture name: 'model_name', backbone used: 'backbone', and training dataset used: 'dataset' to get the respective checkpoint. We term each triplet as a semantic segmentation method. In Table 2, we provide a comprehensive look-up table for all 'model_name' and 'dataset' pairs for which trained checkpoints are available in SEMSEGBENCH.

Table 2: An Overview of all the semantic segmentation methods used in the benchmark in this work made using SEMSEGBENCH. Each of the mentioned backbones has been evaluated using each of the architectures and datasets mentioned in the row in this table.

| Backbone | Architecture | Datasets | Time Proposed (yyyy-mm-dd) |
|---|---|---|---|
| ResNet101 He et al. (2016) | DeepLabV3 Chen et al. (2017), DeepLabV3+ Chen et al. (2018), Mask2Former Cheng et al. (2022), PSPNet Zhao et al. (2017) | ADE20K, Cityscapes, PASCAL VOC 2012 | 2017-12-05 |
| ResNet18 He et al. (2016) | DeepLabV3 Chen et al. (2017), DeepLabV3+ Chen et al. (2018), PSPNet Zhao et al. (2017) | Cityscapes | 2017-12-05 |
| ResNet50 He et al. (2016) | DeepLabV3 Chen et al. (2017), DeepLabV3+ Chen et al. (2018), Mask2Former Cheng et al. (2022), PSPNet Zhao et al. (2017) | ADE20K, Cityscapes, PASCAL VOC 2012 | 2017-12-05 |
| Swin-Base Liu et al. (2021) | Mask2Former Cheng et al. (2022) | ADE20K, Cityscapes, PASCAL VOC 2012 | 2022-06-15 |
| Swin-Small Liu et al. (2021) | Mask2Former Cheng et al. (2022) | ADE20K, Cityscapes, PASCAL VOC 2012 | 2022-06-15 |
| Swin-Tiny Liu et al. (2021) | Mask2Former Cheng et al. (2022) | ADE20K, Cityscapes, PASCAL VOC 2012 | 2022-06-15 |
| MIT-B0 Xie et al. (2021) | SegFormer Xie et al. (2021) | ADE20K, Cityscapes, PASCAL VOC 2012 | 2021-10-28 |
| MIT-B1 Xie et al. (2021) | SegFormer Xie et al. (2021) | ADE20K, Cityscapes, PASCAL VOC 2012 | 2021-10-28 |
| MIT-B2 Xie et al. (2021) | SegFormer Xie et al. (2021) | ADE20K, Cityscapes, PASCAL VOC 2012 | 2021-10-28 |
| MIT-B3 Xie et al. (2021) | SegFormer Xie et al. (2021) | ADE20K, Cityscapes, PASCAL VOC 2012 | 2021-10-28 |
| MIT-B4 Xie et al. (2021) | SegFormer Xie et al. (2021) | ADE20K, Cityscapes, PASCAL VOC 2012 | 2021-10-28 |
| MIT-B5 Xie et al. (2021) | SegFormer Xie et al. (2021) | ADE20K, Cityscapes, PASCAL VOC 2012 | 2021-10-28 |
| UNet Convolutions | UNet Ronneberger et al. (2015) | Cityscapes | 2015-05-18 |
| BEiT-Base Bao et al. (2021) | UPerNet Xiao et al. (2018a) | ADE20K | 2022-09-03 |
| BEiT-Large Bao et al. (2021) | UPerNet Xiao et al. (2018a) | ADE20K | 2022-09-03 |
| InternImage-Base Wang et al. (2023b) | UPerNet Xiao et al. (2018a) | ADE20K, Cityscapes, PASCAL VOC 2012 | 2023-04-17 |
| InternImage-Huge Wang et al. (2023b) | UPerNet Xiao et al. (2018a) | ADE20K | 2023-04-17 |
| InternImage-Large Wang et al. (2023b) | UPerNet Xiao et al. (2018a) | ADE20K, Cityscapes | 2023-04-17 |
| InternImage-Small Wang et al. (2023b) | UPerNet Xiao et al. (2018a) | ADE20K, Cityscapes, PASCAL VOC 2012 | 2023-04-17 |
| InternImage-Tiny Wang et al. (2023b) | UPerNet Xiao et al. (2018a) | ADE20K, Cityscapes, PASCAL VOC 2012 | 2023-04-17 |
| InternImage-XLarge Wang et al. (2023b) | UPerNet Xiao et al. (2018a) | ADE20K, Cityscapes | 2023-04-17 |

# G  SEMSEGBENCH USAGE DETAILS

Following, we provide a detailed description of the evaluation functions and their arguments provided in SEMSEGBENCH. The codebase is available at: `https://anonymous.4open.science/r/benchmarking_reliability_generalization/semantic_segmentation/README.md`.

### G.1 ADVERSARIAL ATTACKS

To evaluate a model for a given dataset on an attack, the following lines of code are required.

```python
from semsegbench.evals import evaluate
model, results = evaluate(
 model_name='upernet', backbone='InterImage-H',
 dataset='ADE20k', retrieve_existing=True,
 threat_config='config.yml')
```

Here, the 'config.yml' contains the configuration for the threat model, for example, when the threat model is a PGD attack, 'config.yml' could contain 'threat_model=*"PGD"*', 'iterations=*20*', 'alpha=*0.01*', 'epsilon=*8*', and 'lp_norm=*"Linf"*'. The argument description is as follows:

- 'model_name' is the name of the semantic segmentation method to be used, given as a string.
- 'dataset' is the name of the dataset to be used, also given as a string.
- 'retrieve_existing' is a boolean flag, which when set to 'True' will retrieve the evaluation from the benchmark if the queried evaluation exists in the benchmark provided by this work, else SEMSEGBENCH will perform the evaluation. If the 'retrieve_existing' boolean flag is set to 'False', then SEMSEGBENCH will perform the evaluation even if the queried evaluation exists in the provided benchmark.
- The 'config.yml' contains the following:
  - 'threat_model' is the name of the adversarial attack to be used, given as a string.
  - 'iterations' are the number of attack iterations, given as an integer.
  - 'epsilon' is the permissible perturbation budget $\epsilon$ given a floating point (float).
  - 'alpha' is the step size of the attack, $\alpha$, given as a floating point (float).
  - 'lp_norm' is the Lipschitz continuity norm ($l_p$-norm) to be used for bounding the perturbation, possible options are 'Linf' and 'L2' given as a string.

### G.2 2D COMMON CORRUPTIONS

To evaluate a model for a given dataset with 2D Common Corruptions, the following lines of code are required.

```python
from semsegbench.evals import evaluate
model, results = evaluate(
 model_name='upernet', backbone='InterImage-H',
 dataset='ADE20k', retrieve_existing=True,
 threat_config='config.yml')
```

Here, the 'config.yml' contains the configuration for the threat model, for example, when the threat model is 2D Common Corruption, 'config.yml' could contain 'threat_model=*"2DCommonCorruption"*', and 'severity=*3*'. Please note, when the 'threat_model' is a common corruption type, SEMSEGBENCH performs evaluations on all corruptions under the respective 'threat_model' and returns the method's performance on each corruption at the requested severity. The argument description is as follows:

- 'model_name' is the name of the semantic segmentation method to be used, given as a string.
- 'dataset' is the name of the dataset to be used also given as a string.
- 'retrieve_existing' is a boolean flag, which when set to 'True' will retrieve the evaluation from the benchmark if the queried evaluation exists in the benchmark provided by this work, else SEMSEGBENCH will perform the evaluation. If the 'retrieve_existing' boolean flag is set to 'False' then SEMSEGBENCH will perform the evaluation even if the queried evaluation exists in the provided benchmark.
- The 'config.yml' contains the following:

- 'threat_model' is the name of the common corruption to be used, given as a string, i.e. '2DCommonCorruption'.
- 'severity' is the severity of the corruption, given as an integer between 1 and 5 (both inclusive).

SEMSEGBENCH supports the following 2D Common Corruption: 'gaussian_noise', shot_noise', 'impulse_noise', 'defocus_blur', 'frosted_glass_blur', 'motion_blur', 'zoom_blur', 'snow', 'frost', 'fog', 'brightness', 'contrast', 'elastic', 'pixelate', 'jpeg'. For the evaluation, SEMSEGBENCH will evaluate the model on the validation images from the respective dataset corrupted using each of the aforementioned corruptions for the given severity, and then report the mean performance over all of them.

## H  DISCUSSING ADVERSARIALLY TRAINED SEMANTIC SEGMENTATION METHODS

Table 3: Comparing the "Robust" PSPNet from Xu et al. (2021) and "Robust" UPerNet Xiao et al. (2018a) with a ConvNeXt-tiny Liu et al. (2022b) backbone against white-box adversarial attacks. Here, same as Xu et al. (2021); Agnihotri et al. (2024b); Croce et al. (2024), $\epsilon = \frac{8}{255}$ and $\alpha$=0.01. These results are obtained from Croce et al. (2024) and Agnihotri et al. (2024b).

| Proposed Method | Training Method | i.i.d. Performance | | Attack Method | 10 Attack Iterations | |
| --- | --- | --- | --- | --- | --- | --- |
| | | mIoU (%) | mAcc (%) | | mIoU (%) | mAcc (%) |
| Robust PSPNet Xu et al. (2021) | No Defense | 76.90 | 84.60 | CosPGD | **0.13** | **0.40** |
| | | | | SegPGD | 1.88 | 5.36 |
| | | | | BIM | 4.14 | 12.22 |
| | SAT Xu et al. (2021) | 74.78 | 83.36 | CosPGD | **17.05** | **38.75** |
| | | | | SegPGD | 20.59 | 43.13 |
| | | | | BIM | 20.67 | 40.05 |
| | DDC-AT Xu et al. (2021) | 75.98 | 84.72 | CosPGD | **23.04** | **41.02** |
| | | | | SegPGD | 25.40 | 42.72 |
| | | | | BIM | 26.90 | 45.27 |
| Robust UPerNet-ConvNeXt-t backbone Croce et al. (2024) | PIR-AT Croce et al. (2024) | 75.20 | 92.70 | **CosAPGD** | 43.73 | 76.36 |
| | | | | SegAPGD | 79.47 | 48.60 |
| | | | | SEA Croce et al. (2024) (100 iterations) | 34.6 | 71.70 |

Some works like (Gu et al., 2022; Agnihotri et al., 2024b; Croce et al., 2024; Xu et al., 2021; Xiao et al., 2018b) have attempted to address the lack of reliability of semantic segmentation methods under adversarial attacks. In Table 3 we bring together some publicly available data on the performance of these methods against SotA adversarial attacks and observe that if an effort is directed towards increasing the reliability of semantic segmentation methods, then this can be achieved using training strategies to some extent. Though, as observed in Table 3, there is still a significant gap between the i.i.d. performance and performance under adversarial attacks, especially SotA attack CosPGD (here CosAPGD, is CosPGD attack, but using APGD (Wong et al., 2020) as an optimizer instead of PGD.), and an ensemble of attacks with 100 attack iterations, Segmentation Ensemble Attack (SEA) as proposed by (Croce et al., 2024). Thus, semantic segmentation methods still need to cover a significant gap to achieve true reliability under attacks.

## I  ADDITIONAL RESULTS

Following, we include additional results from the benchmark made using SEMSEGBENCH.

### I.1  ADVERSARIAL ATTACKS

Here, we report additional results for all adversarial attacks.

#### I.1.1  ITERATIVE ATTACK

Here, we report the evaluations using the PGD, SegPGD, and CosPGD attacks and the correlations between the performance of all considered semantic segmentation methods against these attacks. For $\ell_\infty$-norm bound, perturbation budget $\epsilon = \frac{8}{255}$, and step size $\alpha$=0.01, while for $\ell_2$-norm bound, perturbation budget $\epsilon = 64$ and step size $\alpha$=0.1.

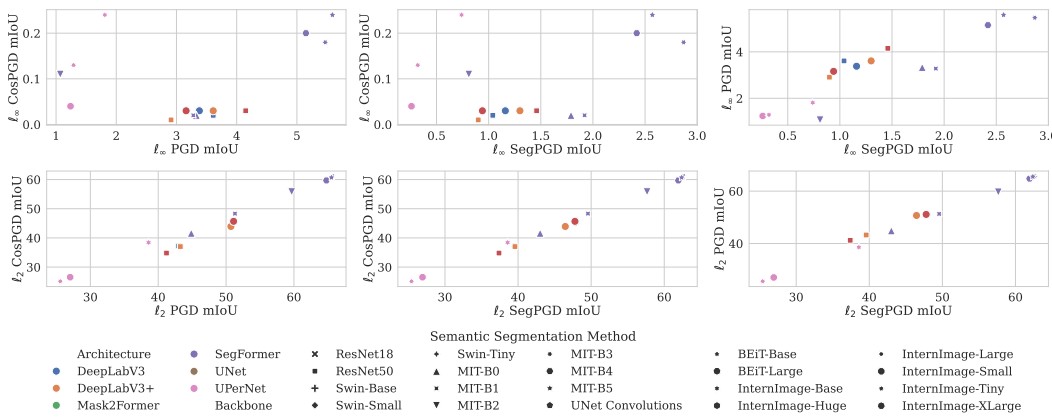

Figure 16: **Dataset used: PASCAL VOC2012**. The correlation in the performance of semantic segmentation methods against different attacks under the $\ell_\infty$-norm and $\ell_2$-norm bounded attacks. The respective axis shows the name of the attack used. Colors are used to show different architectures and marker styles are used to show different backbones used by the semantic segmentation methods. We observe that $\ell_\infty$-norm bounded CosPGD attack is very strong, bringing down the performance of almost all methods to almost 0.0 mIoU, and thus it does not have any observable correlation with other attacks. However, in other cases, there is a strong correlation in the performance of methods under different attacks.

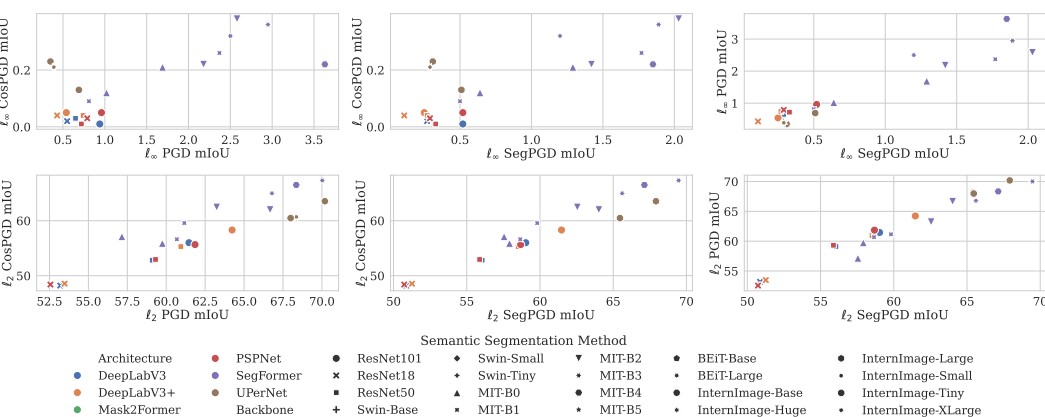

Figure 17: **Dataset used: Cityscapes**. The correlation in the performance of semantic segmentation methods against different attacks under the $\ell_\infty$-norm and $\ell_2$-norm bounded attacks. The respective axis shows the name of the attack used. Colors are used to show different architectures and marker styles are used to show different backbones used by the semantic segmentation methods. We observe that $\ell_\infty$-norm bounded CosPGD attack is very strong, bringing down the performance of almost all methods to almost 0.0 mIoU, and thus it does not have any observable correlation with other attacks. However, in other cases, there is a strong correlation in the performance of methods under different attacks.

We show the correlation between different attacks for PASCAL VOC2012 in Figure 16, Cityscapes in Figure 17, and for ADE20K in Figure 18.

We show individual attack evaluations for PASCAL VOC2012 in Figure 19, Cityscapes in Figure 20, and for ADE20K in Figure 21.

Please note that due to the architectural implementation of Mask2Former (Cheng et al., 2022), it is not possible to get pixel-wise loss for this architecture. Therefore, evaluations using SegPGD and CosPGD for Mask2Former are not possible without substantial changes to the architecture's implementation. To the best of our knowledge, such a change is beyond the scope of this work. Under 20

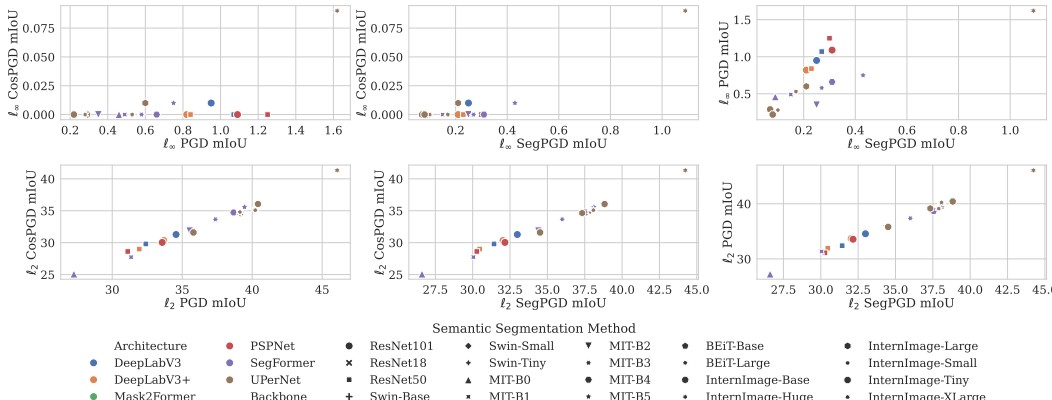

Figure 18: **Dataset used: ADE20K**. The correlation in the performance of semantic segmentation methods against different attacks under the $\ell_\infty$-norm and $\ell_2$-norm bounded attacks. The respective axis shows the name of the attack used. Colors are used to show different architectures and marker styles are used to show different backbones used by the semantic segmentation methods. We observe that $\ell_\infty$-norm bounded CosPGD attack is very strong, bringing down the performance of almost all methods to almost 0.0 mIoU, and thus it does not have any observable correlation with other attacks. However, in other cases, there is a strong correlation in the performance of methods under different attacks.

attack iterations, it appears that Mask2Former is marginally more robust than other methods under PGD attack. To validate this, we perform attacks on Mask2Former and SegFormer with a different setting. Under 40 iterations $\epsilon = \frac{2}{255}$ $\ell_\infty$-norm PGD attack using ADE20K, SegFormer MIT-B1 has 3.8% mIoU and 6.73% mAcc while Mask2Former has 0.35% mIoU and 0.97% mAcc. Thus, Mask2Former is merely harder to attack. One explanation for this phenomenon is that the sparsity in the feature representation by Mask2Former due to masking of the attention heads is inherently increasing the model's robustness to some extent; this explanation is supported by findings from (Liao et al., 2022; Chen et al., 2022; Peng et al., 2023).

### I.2 2D COMMON CORRUPTIONS

Following we provide an overview of the performance of all the semantic segmentation methods over all of the common corruptions, for PASCAL VOC2012 in Figure 22, for Cityscapes in Figure 23, and for ADE20K in Figure 24.

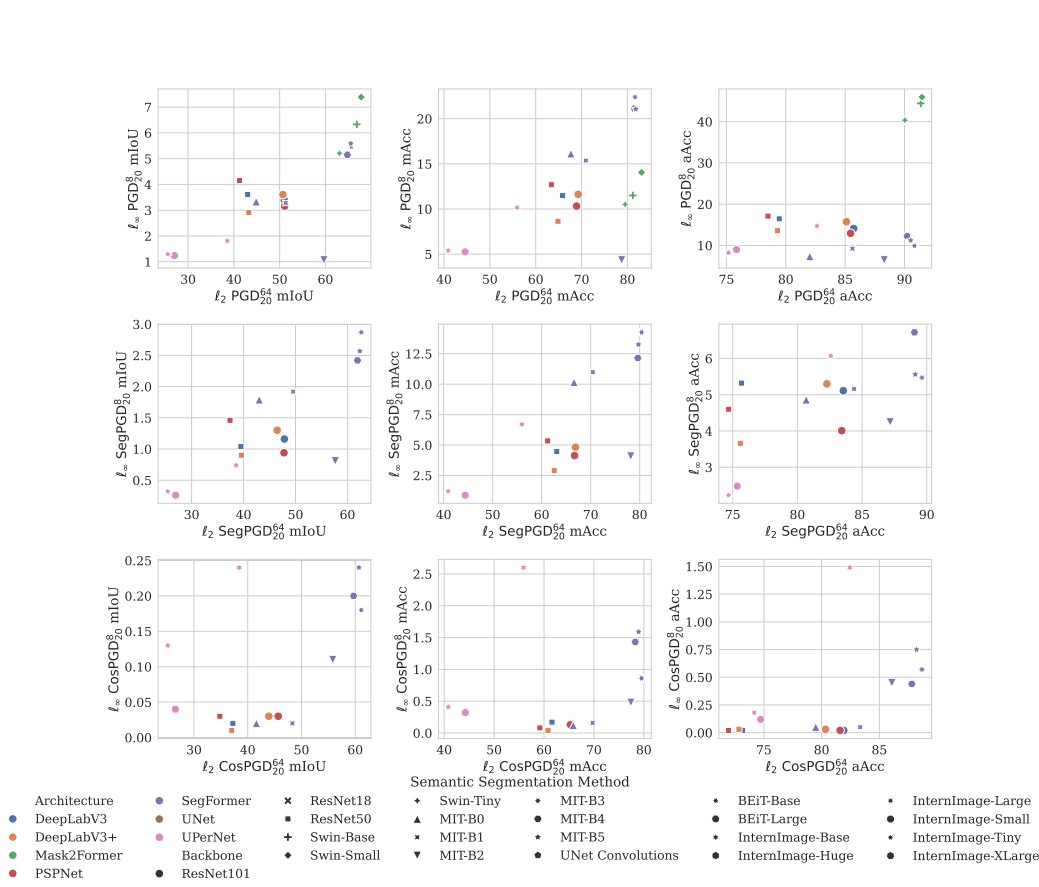

Figure 19: **Dataset used: PASCAL VOC2012.** Reliability of semantic segmentation methods against individual attacks. TOP: PGD, MIDDLE: SegPGD and BOTTOM: CosPGD, constrained under $\ell_\infty$-norm (y-axis) and the $\ell_2$-norm (x-axis).

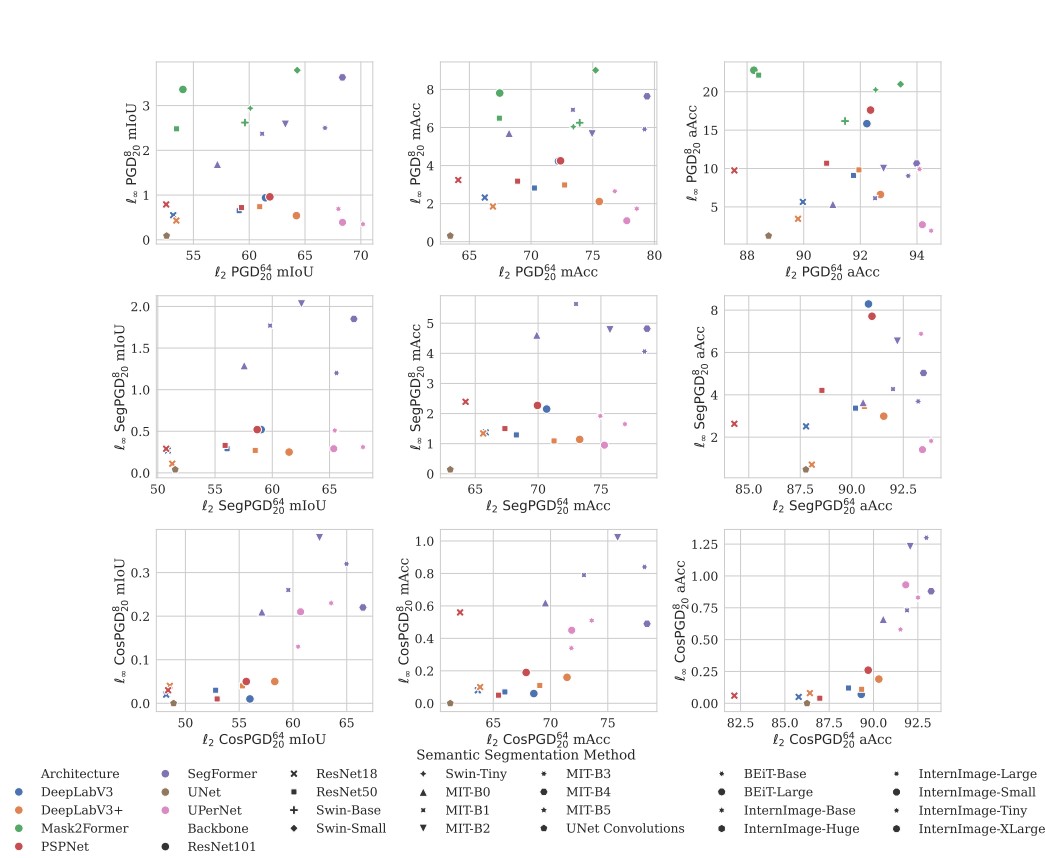

Figure 20: **Dataset used: Cityscapes.** Reliability of semantic segmentation methods against individual attacks. TOP: PGD, MIDDLE: SegPGD and BOTTOM: CosPGD, constrained under $\ell_\infty$-norm (y-axis) and the $\ell_2$-norm (x-axis).

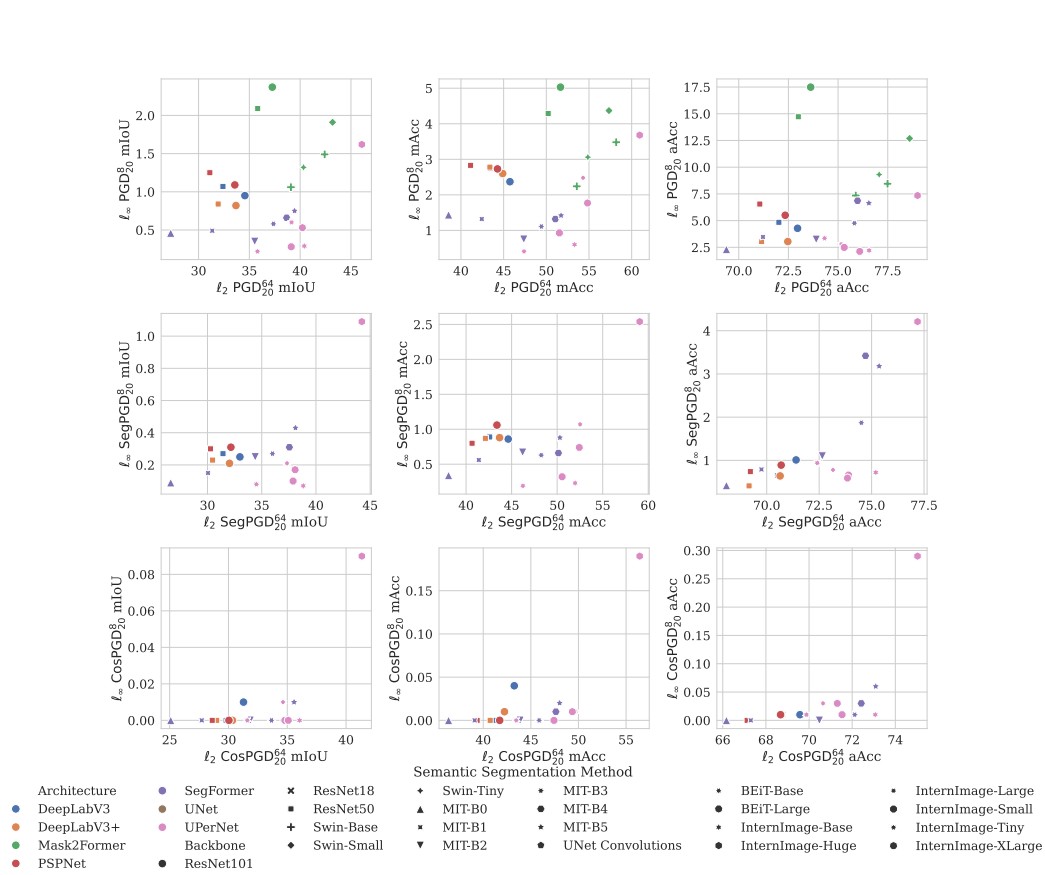

Figure 21: **Dataset used: ADE20K.** Reliability of semantic segmentation methods against individual attacks. TOP: PGD, MIDDLE: SegPGD and BOTTOM: CosPGD, constrained under $\ell_\infty$-norm (y-axis) and the $\ell_2$-norm (x-axis).

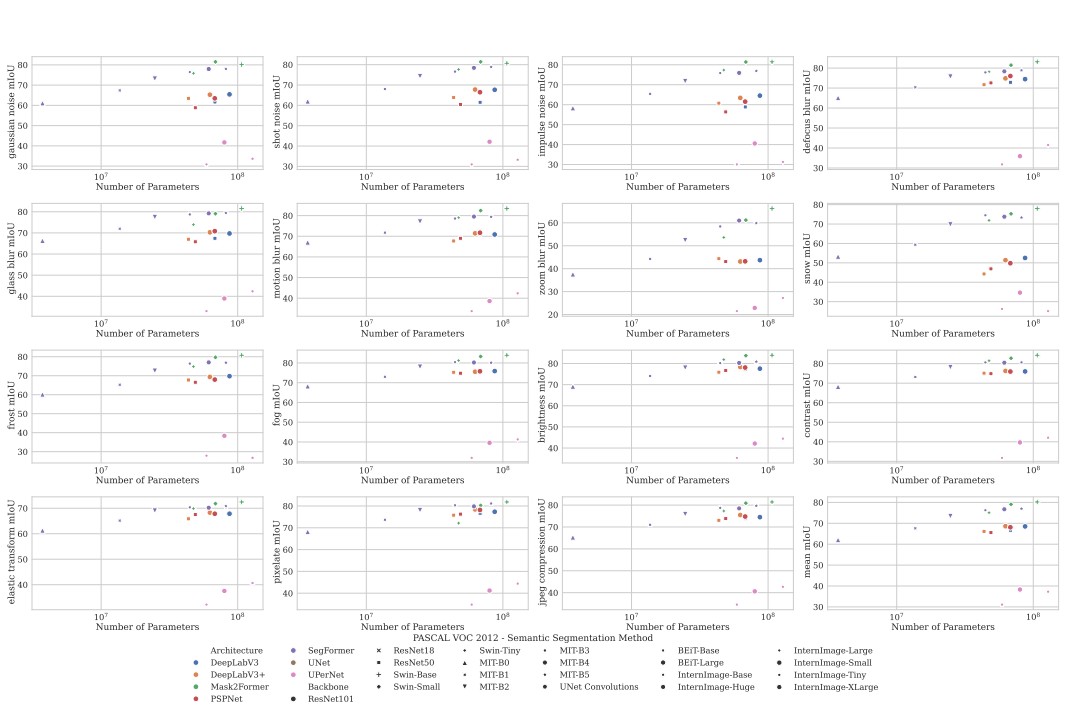

Figure 22: **Dataset used: PASCAL VOC2012.** The correlation in the performance of semantic segmentation methods against different 2D Common Corruptions. The respective axis shows the name of the common corruption used. Colors are used to show different architectures and marker styles are used to show different backbones used by the semantic segmentation methods. For the limited PASCAL VOC2012 evaluations we observe some correlation between the number of learnable parameters and the performance against common corruptions, however, more evaluations (more publicly available checkpoints) are required for a meaningful analysis.

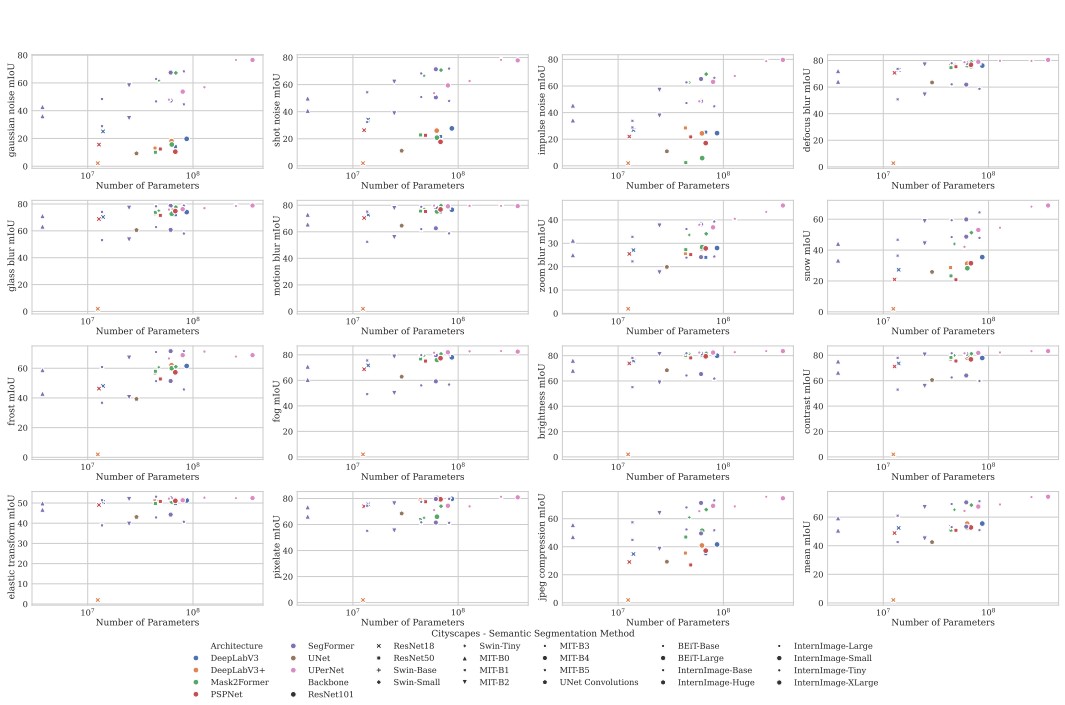

Figure 23: **Dataset used: Cityscapes**. The correlation in the performance of semantic segmentation methods against different 2D Common Corruptions. The respective axis shows the name of the common corruption used. Colors are used to show different architectures and marker styles are used to show different backbones used by the semantic segmentation methods. Except for DeepLabV3+ with a ResNet18 backbone, most other methods show a weak positive correlation between the number of learnable parameters used by a method and its performance against most of the common corruption. Multiple occurrences of an Architecture and Backbone pair are due to their evaluations being performed at two different crop sizes i.e. 512×512, and 512×1024.

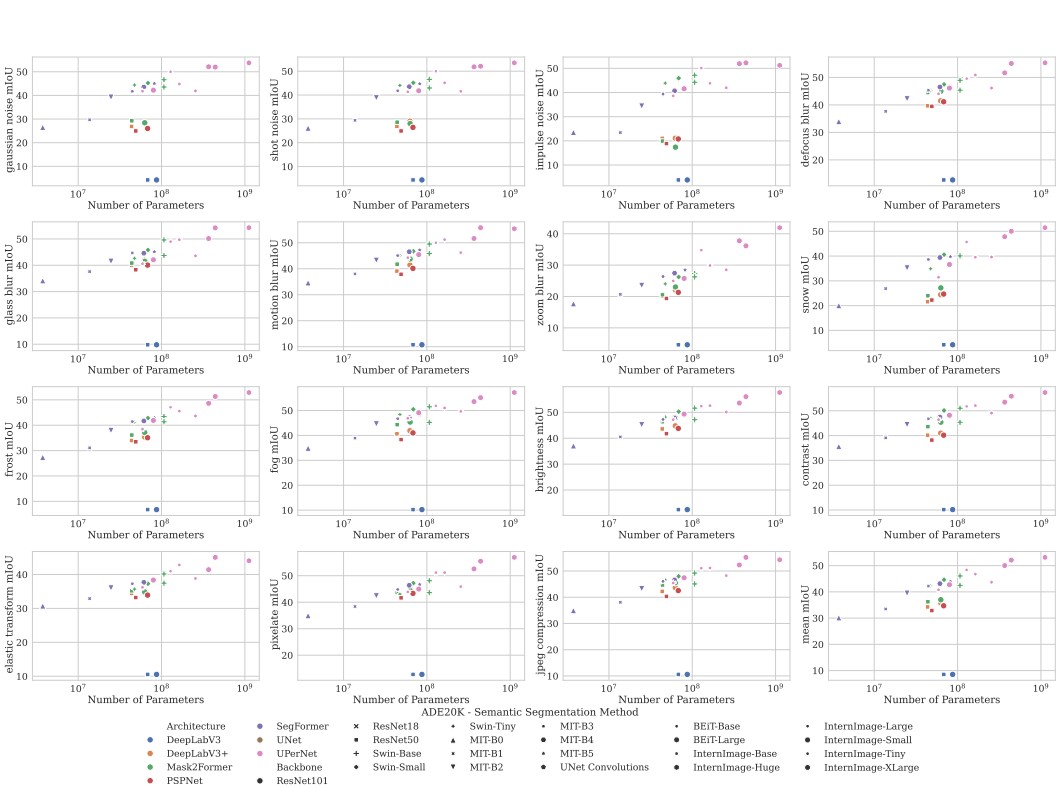

Figure 24: **Dataset used: ADE20K**. The correlation in the performance of semantic segmentation methods against different 2D Common Corruptions. The respective axis shows the name of the common corruption used. Colors are used to show different architectures and marker styles are used to show different backbones used by the semantic segmentation methods. Except for DeepLabV3, all other methods show some positive correlation between the number of learnable parameters used by a method and its performance against any common corruption.

# Additional Details on the Benchmarking using DETECBENCH and Evaluations (All Benchmarking Results):

TABLE OF CONTENT

The supplementary material covers the following information:

## J DATASET DETAILS

DETECBENCH supports a total of two object detection datasets. Following, we describe these datasets in detail.

### J.1 MS-COCO

The MS-COCO dataset (Lin et al., 2014): Common Objects in Context is a large-scale image recognition dataset proposed for object detection, semantic segmentation, and captioning tasks. For the tasks considered in this work, we use the 2017 version, which has 120k labeled images with annotations for 80 different object categories split into 115k for training and 5k for validation.

## J.2    PASCAL VOC

The PASCAL VOC 2007 (Everingham et al., 2010) dataset has 20 object classes and a total of 9963 images split 50-50 into training and testing.

## K    IMPLEMENTATION DETAILS OF THE BENCHMARK

Following we provide details regarding the experiments done for creating the benchmark used in the analysis.

**Compute Resources.**    Most experiments were done on a single 40 GB NVIDIA Tesla V100 GPU each, however, some recently proposed models with large backbones are more compute-intensive, and thus 80GB NVIDIA A100 GPUs or NVIDIA H100 were used for these models, a single GPU for each experiment. Training some of the architectures with large backbones required using two to four GPUs in parallel.

**Datasets Used.**    Performing adversarial attacks and OOD robustness evaluations are very expensive and compute-intensive. Thus, performing training for all model-dataset pairs is not possible given the limited computing resources at our disposal. Thus, we benchmark publicly available models while training a few on the MS-COCO dataset.

**Metrics Calculation.**    In Sec. 3 we introduce two new metrics for better understanding our analysis, given the large scale of the benchmark created. For calculating $\mathrm{ReM}$ values we used the BIM, and PGD attacks with step size $\alpha$=0.01, perturbation budget $\epsilon = \frac{8}{255}$ under the $\ell_\infty$-norm bound, as non-targeted attacks. We use 20 attack iterations for calculating $\mathrm{ReM}$ as we observe in $Appendix\ O$, that even after just 10 attack iterations, the attacks seem to saturate, and as shown by (Agnihotri et al., 2024b) and (Schmalfuss et al., 2022b), 20 iterations are enough to optimize an attack to truly understand the performance of the attacked method. For calculating $\mathrm{GAM}$, we use all 15 2D Common Corruptions: 'Gaussian Noise', Shot Noise', 'Impulse Noise', 'Defocus Blur', 'Frosted Glass Blur', 'Motion Blur', 'Zoom Blur', 'Snow', 'Frost', 'Fog', 'Brightness', 'Contrast', 'Elastic Transform', 'Pixelate', 'JPEG Compression', and eight 3D Common Corruptions: 'Color Quantization', 'Far Focus', 'Fog 3D', 'ISO Noise', 'Low Light', 'Near Focus', 'XY Motion Blur', and 'Z Motion Blur'. We could not use 'h265 crf' and 'h265 abr' as computing these for Conditional-DETR and Co-DETR was computationally infeasible given the limited compute resources. For the other methods we compute evaluations against 'h265 crf' and 'h265 abr', however, for fairness, we do not use them when calculating the $\mathrm{GAM}$ values. All the common corruptions are at severity 3.

**Calculating the mAP.**    mAP is the mean Average Precision calculated over the entire evaluation set, this is the primary metric used for object detection method evaluations across datasets. Additionally, for MS-COCO, we also record the $\mathrm{mAP_{small}}$ ($\mathrm{mAP_s}$) for small sized objects, $\mathrm{mAP_{medium}}$ ($\mathrm{mAP_m}$) for medium sized objects, and $\mathrm{mAP_{large}}$ ($\mathrm{mAP_l}$) for large sized objects. Moreover, we also capture $\mathrm{mAP_{50}}$ and $\mathrm{mAP_{75}}$ for 50% and 75% mIoU with the ground truth bounding boxes.

**Models Used.**    All available checkpoints, as shown in Tab. 4 for the MS-COCO and PASCAL VOC dataset were used for creating the benchmark. Our evaluations include old DL-based object detection methods Faster-RCNN (Ren, 2015), as well as recent state-of-the-art methods like MM Grounding DINO (Zhao et al., 2024).

## L    DESCRIPTION OF DETECBENCH

Following, we describe the benchmarking tool, DETECBENCH. It is built using mmdetection (Chen et al., 2019), and supports almost all prominent object detection method architectures and backbones and 3 distinct datasets, namely MS-COCO (Lin et al., 2014), PASCAL VOC 2007 (Everingham et al., 2010), LVIS (Gupta et al., 2019) datasets (please refer Appendix J for additional details on the datasets). DETECBENCH goes beyond mmdetection as it enables training and evaluations on all

aforementioned datasets including evaluations using adversarial attacks such as BIM (Kurakin et al., 2018), PGD (Kurakin et al., 2017), and FGSM (Goodfellow et al., 2015), under various lipshitz ($l_p$) norm bounds.

Additionally, it enables evaluations for Out-of-Distribution (OOD) robustness by corrupting the inference samples using 2D Common Corruptions (Hendrycks & Dietterich, 2019) and 3D Common Corruptions (Kar et al., 2022).

We follow the nomenclature set by RobustBench (Croce et al., 2021) and use "threat_model" to define the kind of evaluation to be performed. When "threat_model" is defined to be "None", the evaluation is performed on unperturbed and unaltered images, if the "threat_model" is defined to be an adversarial attack, for example "FGSM", "PGD" or "BIM", then DETECBENCH performs an adversarial attack using the user-defined parameters. We elaborate on this in Appendix L.1. Whereas, if "threat_model" is defined to be "2DCommonCorruptions" or "3DCommonCorruptions", the DETECBENCH performs evaluations after perturbing the images with 2D Common Corruptions and 3D Common Corruptions respectively. We elaborate on this in Appendix L.2.

If the queried evaluation already exists in the benchmark provided by this work, then DETECBENCH simply retrieves the evaluations, thus saving computation.

## L.1 ADVERSARIAL ATTACKS

Due to significant similarity, most of the text here has been adapted from (Agnihotri et al., 2025). DETECBENCH enables the use of all the attacks mentioned in Sec. 2 to help users better study the reliability of their object detection methods. We choose to specifically include these white-box adversarial attacks as they serve as the common benchmark for adversarial attacks in classification literature (FGSM, BIM, PGD) for testing the reliability of methods. These attacks are currently designed to be both *Non-targeted* when they simply fool the model into making incorrect predictions, irrespective of what the model eventually predicts, and *Targeted*, when they fool the model into making specific incorrect predictions. Following, we discuss these attacks in detail and highlight their key differences.

**FGSM.** Assuming a non-targeted attack, given a model $f_\theta$ and an unperturbed input sample $\boldsymbol{X}^{\mathrm{clean}}$ and ground truth label $\boldsymbol{Y}$, FGSM attack adds noise $\delta$ to $\boldsymbol{X}^{\mathrm{clean}}$ as follows,

$$\boldsymbol{X}^{\mathrm{adv}} = \boldsymbol{X}^{\mathrm{clean}} + \alpha \cdot \mathrm{sign}\nabla_{\boldsymbol{X}^{\mathrm{clean}}} L(f_\theta(\boldsymbol{X}^{\mathrm{clean}}), \boldsymbol{Y}), \tag{11}$$

$$\delta = \phi^\epsilon(\boldsymbol{X}^{\mathrm{adv}} - \boldsymbol{X}^{\mathrm{clean}}), \tag{12}$$

$$\boldsymbol{X}^{\mathrm{adv}} = \phi^r(\boldsymbol{X}^{\mathrm{clean}} + \delta). \tag{13}$$

Here, $L(\cdot)$ is the loss function (differentiable at least once) which calculates the loss between the model prediction and ground truth, $\boldsymbol{Y}$. $\alpha$ is a small value of $\epsilon$ that decides the size of the step to be taken in the direction of the gradient of the loss w.r.t. the input image, which leads to the input sample being perturbed such that the loss increases. $\boldsymbol{X}^{\mathrm{adv}}$ is the adversarial sample obtained after perturbing $\boldsymbol{X}^{\mathrm{clean}}$. To make sure that the perturbed sample is semantically indistinguishable from the unperturbed clean sample to the human eye, steps from Eq. (12) and Eq. (13) are performed. Here, function $\phi^\epsilon$ is clipping the $\delta$ in $\epsilon$-ball for $\ell_\infty$-norm bounded attacks or the $\epsilon$-projection in other $l_p$-norm bounded attacks, complying with the $\ell_\infty$-norm or other $l_p$-norm constraints, respectively. While function $\phi^r$ clips the perturbed sample ensuring that it is still within the valid input space. FGSM, as proposed, is a single step attack. For targeted attacks, $\boldsymbol{Y}$ is the target and $\alpha$ is multiplied by -1 so that a step is taken to minimize the loss between the model's prediction and the target prediction.

**BIM.** This is the direct extension of FGSM into an iterative attack method. In FGSM, $\boldsymbol{X}^{\mathrm{clean}}$ was perturbed just once. While in BIM, $\boldsymbol{X}^{\mathrm{clean}}$ is perturbed iteratively for time steps $t \in [0, \boldsymbol{T}]$, such

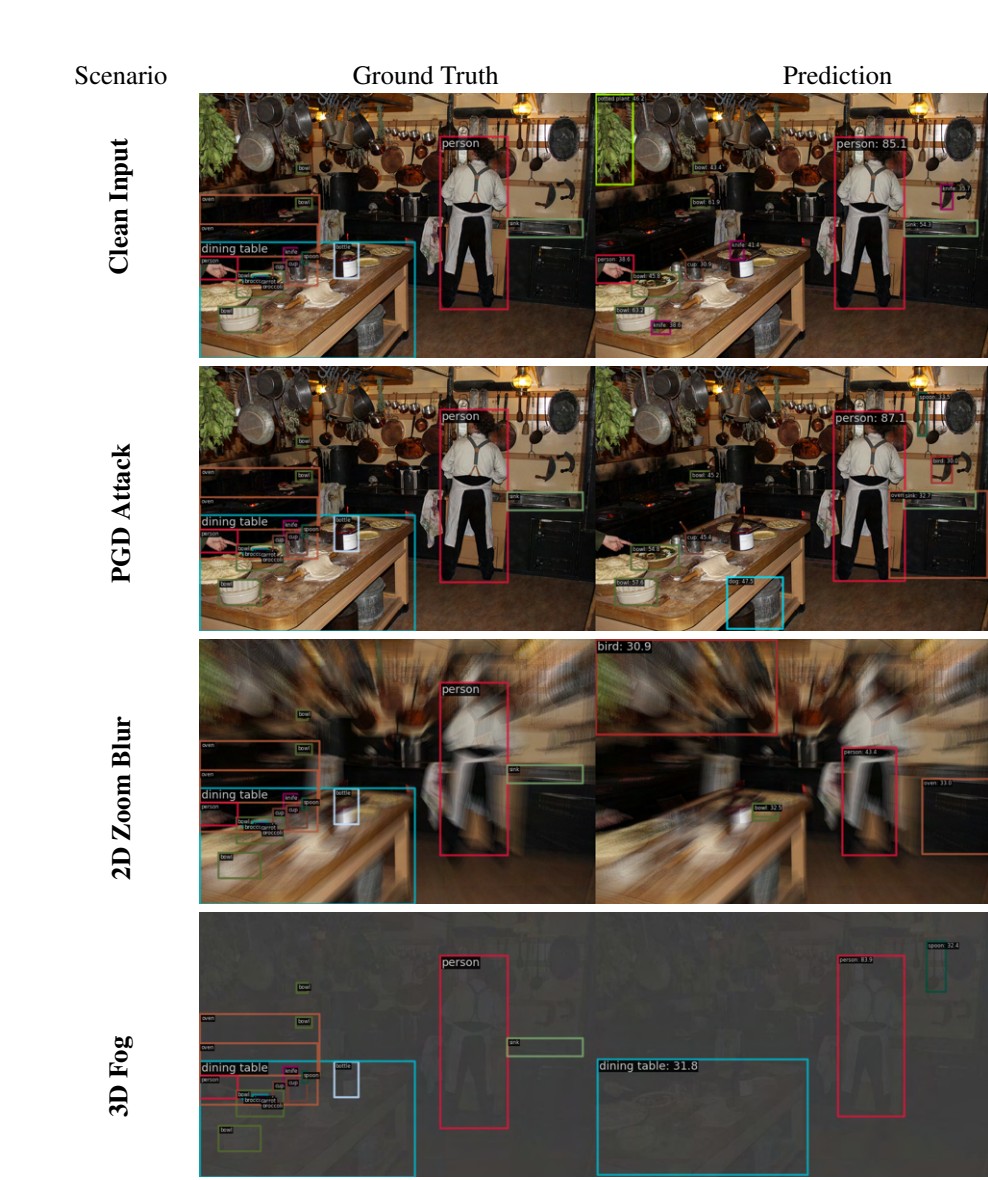

Figure 25: Using **DeFormable-DETR with a ConvNeXt-small Backbone** and **MS-COCO dataset** we show for an example input image the changes in prediction under PGD attack, under Zoom Blur corruption from 2D Common Corruptions and under Fog corruption from 3D Common Corruptions. In each image, the **left image is the ground truth** bounding box and class prediction, while the **right image is the predicted bounding box and class**. We observe that under each threat model, the predictions are incorrect when compared to the ground truth. While the common corruptions cause the model to incorrectly predict the bounding boxes and miss most objects, the PGD attack is fooling the model into hallucinating objects that do not exist in the input image. This is a critical threat for the real-world deployment of object detection methods.

that $t \in \mathbb{Z}^+$, where $\boldsymbol{T}$ are the total number of permissible attack iterations. This changes the steps of the attack from FGSM to the following,

$$\boldsymbol{X}^{\mathrm{adv}_{t+1}} = \boldsymbol{X}^{\mathrm{adv}_t} + \alpha \cdot \mathrm{sign} \nabla_{\boldsymbol{X}^{\mathrm{adv}_t}} L(f_\theta(\boldsymbol{X}^{\mathrm{adv}_t}), \boldsymbol{Y}), \quad (14)$$

$$\delta = \phi^\epsilon(\boldsymbol{X}^{\mathrm{adv}_{t+1}} - \boldsymbol{X}^{\mathrm{clean}}), \quad (15)$$

$$\boldsymbol{X}^{\mathrm{adv}_{t+1}} = \phi^r(\boldsymbol{X}^{\mathrm{clean}} + \delta). \quad (16)$$

Here, at $t$=0, $\boldsymbol{X}^{\mathrm{adv}_t} = \boldsymbol{X}^{\mathrm{clean}}$.

**PGD.**  Since in BIM, the initial prediction always started from $\boldsymbol{X}^{\mathrm{clean}}$, the attack required a significant amount of steps to optimize the adversarial noise and yet it was not guaranteed that in the permissible $\epsilon$-bound, $\boldsymbol{X}^{\mathrm{adv}_{t+1}}$ was far from $\boldsymbol{X}^{\mathrm{clean}}$. Thus, PGD proposed introducing stochasticity to ensure random starting points for attack optimization. They achieved this by perturbing $\boldsymbol{X}^{\mathrm{clean}}$ with $\mathcal{U}(-\epsilon, \epsilon)$, a uniform distribution in $[-\epsilon, \epsilon]$, before making the first prediction, such that, at $t$=0

$$\boldsymbol{X}^{adv_t} = \phi^r(\boldsymbol{X}^{clean} + \mathcal{U}(-\epsilon, \epsilon)). \quad (17)$$

Fig. 25, shows adversarial examples created using an adversarial attack and how it affects the model's predictions.

## L.2  OUT-OF-DISTRIBUTION ROBUSTNESS

Due to significant similarity, most of the text here has been adapted from (Agnihotri et al., 2025). While adversarial attacks help explore vulnerabilities of inefficient feature representations learned by a model, another important aspect of reliability is generalization ability. Especially, generalization to previously unseen samples or samples from significantly shifted distributions compared to the distribution of the samples seen while learning model parameters. As one cannot cover all possible scenarios during model training, a certain degree of generalization ability is expected from models. However, multiple works (Hendrycks & Dietterich, 2019; Kar et al., 2022; Hoffmann et al., 2021) showed that models are surprisingly less robust to distribution shifts, even those that can be caused by commonly occurring phenomena such as weather changes, lighting changes, etc. This makes the study of Out-of-Distribution (OOD) robustness an interesting avenue for research. Thus, to facilitate the study of robustness to such commonly occurring corruptions, DETECBENCH enables evaluating against prominent image corruption methods. Following, we describe these methods in detail.

**2D Common Corruptions.**  (Hendrycks & Dietterich, 2019) propose introducing distribution shift in the input samples by perturbing images with a total of 15 synthetic corruptions that could occur in the real world. These corruptions include weather phenomena such as fog, and frost, digital corruptions such as jpeg compression, pixelation, and different kinds of blurs like motion, and zoom blur, and noise corruptions such as Gaussian and shot noise amongst others corruption types. Each of these corruptions can perturb the image at 5 different severity levels between 1 and 5. The final performance of the model is the mean of the model's performance on all the corruptions, such that every corruption is used to perturb each image in the evaluation dataset. Since these corruptions are applied to a 2D image, they are collectively termed 2D Common Corruptions.

**3D Common Corruptions.**  Since the real world is 3D, (Kar et al., 2022) extend 2D Common Corruptions to formulate more realistic-looking corruptions by leveraging depth information (synthetic depth information when real depth is not readily available) and luminescence angles. They name these image corruptions as 3D Common Corruptions. Fig. 25, shows examples of MS-COCO image corrupted using 2D Common Corruption and 3D Common Corruption, and how these affect the model's prediction, making the model predict incorrect bounding boxes.

## M  MODEL ZOO

The trained checkpoints for all models available in DETECBENCH can be obtained using the following lines of code:

```
from detecbench import load_model
model = load_model(model_folder='models/DINO_Swin-L')
```

Users need to specify the path to the model folder: 'model_folder'. This path should contain the weights and configuration for the model to be loaded. In Table 4, we provide a comprehensive look-up table for all configurations and weights for architecture and dataset pairs for which trained checkpoints are available in DETECBENCH.

Table 4: An Overview of all the object detection methods used in the benchmark in this work made using DETECBENCH.

| Architecture | Backbones | Time Proposed |
|---|---|---|
| **MS-COCO Dataset** | | |
| ATSS Zhang et al. (2020b) | ResNet101 He et al. (2016) | 20.06.2020 |
| Cascade Mask R-CNN Cai & Vasconcelos (2019) | ResNet101 He et al. (2016), ResNeXt101 Xie et al. (2017b) | 24.06.2019 |
| Cascade R-CNN Cai & Vasconcelos (2019) | ResNet101 He et al. (2016), ResNeXt101 Xie et al. (2017b), ConvNeXt-S Liu et al. (2022b), Swin-S Liu et al. (2021) | 24.06.2019 |
| CenterNet Duan et al. (2019) | ResNet18 He et al. (2016) | 25.04.2019 |
| Co-DETR Zong et al. (2023) | ResNet50 He et al. (2016), Swin-L Liu et al. (2021) | 10.08.2023 |
| Conditional DETR Meng et al. (2021) | ResNet50 He et al. (2016) | 29.09.2023 |
| DAB-DETR Liu et al. (2022a) | ResNet50 He et al. (2016) | 30.03.2022 |
| DDOD-ATSS Chen et al. (2021) | ResNet50 He et al. (2016) | 27.07.2021 |
| DDQ DETR Zhang et al. (2023) | Swin-L Liu et al. (2021), ResNet50 He et al. (2016) | 05.07.2023 |
| Deformable DETR Zhu et al. (2021) | ResNet50 He et al. (2016), ConvNeXt-S Liu et al. (2022b), Swin-S Liu et al. (2021) | 18.03.2021 |
| DETR Carion et al. (2020) | ResNet50 He et al. (2016) | 28.05.2020 |
| DINO Caron et al. (2021) | ResNet50 He et al. (2016), Swin-L Liu et al. (2021) | 11.07.2022 |
| Double Heads Wu et al. (2020) | ResNet50 He et al. (2016) | 02.04.2020 |
| Dynamic R-CNN Zhang et al. (2020a) | ResNet50 He et al. (2016) | 26.07.2020 |
| Faster R-CNN Ren (2015) | ResNet101 He et al. (2016), ResNeXt101 Xie et al. (2017b) | 06.01.2016 |
| FCOS Tian et al. (2022) | ResNeXt101 Xie et al. (2017b) | 20.08.2019 |
| FoveaBox Kong et al. (2020) | ResNet101 Xie et al. (2017b) | 16.07.2020 |
| FreeAnchor Zhang et al. (2019) | ResNet101 He et al. (2016), ResNeXt101 Xie et al. (2017b) | 12.11.2019 |
| FSAF Zhu et al. (2019) | ResNet101 He et al. (2016), ResNeXt101 Xie et al. (2017b) | 02.03.2019 |
| GA-Faster R-CNN Wang et al. (2019) | ResNeXt101 Xie et al. (2017b) | 12.04.2019 |
| GA-RetinaNet Wang et al. (2019) | ResNeXt101 Xie et al. (2017b) | 12.04.2019 |
| GLIP-L Li et al. (2022c) | Swin-L Liu et al. (2021) | 17.06.2022 |
| Grid R-CNN Lu et al. (2019) | ResNet101 He et al. (2016), ResNeXt101 Xie et al. (2017b) | 29.11.2018 |
| Libra R-CNN Pang et al. (2019) | ResNet101 He et al. (2016), ResNeXt101 Xie et al. (2017b) | 04.04.2019 |
| PAA Kim & Lee (2020) | ResNet101 He et al. (2016), ConvNeXt-S Liu et al. (2022b), Swin-S Liu et al. (2021) | 05.09.2020 |
| RepPoints Yang et al. (2019) | ResNet101 He et al. (2016), ResNeXt101 Xie et al. (2017b) | 19.08.2019 |
| RetinaNet Lin (2017) | ResNet101 He et al. (2016), ResNeXt101 Xie et al. (2017b) | 07.02.2018 |
| RTMDet-l Lyu et al. (2022) | ConvNeXt-B Liu et al. (2022b), Swin-B Liu et al. (2021) | 16.12.2022 |
| SABL Cascade R-CNN Wang et al. (2020) | ResNet101 He et al. (2016) | 26.08.2020 |
| SABL Faster R-CNN Wang et al. (2020) | ResNet101 He et al. (2016) | 26.08.2020 |
| SABL RetinaNet Wang et al. (2020) | ResNet101 He et al. (2016) | 26.08.2020 |
| Sparse R-CNN Sun et al. (2021) | ResNet101 He et al. (2016), ConvNeXt-S Liu et al. (2022b), Swin-S Liu et al. (2021) | 26.04.2021 |
| TOOD Feng et al. (2021) | ResNet101 He et al. (2016), ResNeXt101 Xie et al. (2017b), ConvNeXt-S Liu et al. (2022b), Swin-S Liu et al. (2021) | 28.08.2021 |
| VarifocalNet Zhang et al. (2021) | ResNet101 He et al. (2016), ResNeXt101 Xie et al. (2017b) | 04.03.2021 |
| MM Grounding DINO Zhao et al. (2024) | Swin-L Liu et al. (2021) | 05.01.2024 |
| **PASCAL VOC Dataset** | | |
| Faster R-CNN Ren (2015) | ResNet50 He et al. (2016) | 06.01.2016 |
| RetinaNet Lin (2017) | ResNet50 He et al. (2016) | 07.02.2018 |

# N    DETECBENCH USAGE DETAILS

Following we provide a detailed description of the evaluation functions and their arguments provided in DETECBENCH.

The codebase is available at: `https://anonymous.4open.science/r/benchmarking_reliability_generalization/object_detection/README.md`.

## N.1    ADVERSARIAL ATTACKS

To evaluate a model for a given dataset, in an attack, the following lines of code are required.

```
from detecbench import attacks, evaluate
pgd = attacks.PGD(
    epsilon = 8,
    alpha = 2.55,
    steps = 20,
    norm = "inf",
    target = False,
    random_start = True,
)
evaluate(task=pgd, model_folder="./models/DINO_Swin-L", log_dir =
↪   "./logs", wandb_project = None, wandb_entity = None)
```

Here, the 'model_folder' accepts the configuration and weights for the model, for example, when the model is DINO_Swin-L, 'model_folder' should be passed a folder that contains the configuration and the weights for the model, the dataset can be controlled with this as well, that is by providing the name of the dataset and the data root folder in the configuration file. Here, the threat model=*"PGD"*, 'steps=*20*', 'alpha=*2.55*', 'epsilon=*8*', and 'norm=*"inf"*'. 'random_start=*False*' leads to BIM attack and 'random_start=*True*' leads to a PGD attack. We additionally provide two types of logging, either using 'logger' and/or 'wandb'. The argument description is as follows:

- 'model_folder' is the directory that contains the model weights and configurations.
- 'dataset' is the name of the dataset to be used, also given as a string.
- The arguments for *attack.PGD()* contains the following:
    - 'steps' is the number of attack iterations, given as an integer.
    - 'epsilon' is the permissible perturbation budget $\epsilon$ given a floating point (float).
    - 'alpha' is the step size of the attack, $\alpha$, given as a floating point (float).
    - 'norm' is the Lipschitz continuity norm ($l_p$-norm) to be used for bounding the perturbation, possible options are 'inf' and 'two' given as a string.
    - 'target' is false by default, but to do targeted attacks, either the user can set 'target'=True, to use the default class label 42 as the target, or can pass an integer for the class label to be used as the target image-wide, or can pass the path (as string) to a specific tensor to be used as a target.

Please refer to our code, `https://anonymous.4open.science/r/benchmarking_` `reliability_generalization/object_detection/README.md`, for additional attack settings.

### N.2   2D COMMON CORRUPTIONS

To evaluate a model for a given dataset, with 2D Common Corruptions, the following lines of code are required.

```
from detecbench import corruptions, evaluate
cc_contrast = corruptions.CommonCorruption(name="contrast", severity=3)
evaluate(task=cc_contrast, model_folder="./models/RetinaNet_R-101-FPN",
↪    log_dir = "./logs", wandb_project = None, wandb_entity = None)
```

Here, the 'model_folder' contains the configuration and weights for the model for a given dataset. Please note, the 'threat model' is a common corruption type, for example, here 'contrast'. To use DETECBENCH to perform evaluations on all corruptions under the respective 'threat model', use 'all' as the '*name*'. We additionally provide two types of logging, either using 'logger' and/or 'wandb'.

DETECBENCH supports the following 2D Common Corruption: 'gaussian_noise', shot_noise', 'impulse_noise', 'defocus_blur', 'frosted_glass_blur', 'motion_blur', 'zoom_blur', 'snow', 'frost', 'fog', 'brightness', 'contrast', 'elastic', 'pixelate', 'jpeg'. For the evaluation, DETECBENCH will evaluate the model on the validation images from the respective dataset corrupted using each of the aforementioned corruptions for the given severity.

### N.3   3D COMMON CORRUPTIONS

To evaluate a model for a given dataset, with 3D Common Corruptions, the following lines of code are required.

```
from detecbench import corruptions, evaluate

cc3d_near_focus = corruptions.CommonCorruption3d(name="near_focus",
↪    severity=3)
evaluate(task=cc3d_near_focus,
↪    model_folder="./models/RetinaNet_R-101-FPN", log_dir = "./logs",
↪    wandb_project = None, wandb_entity = None)
```

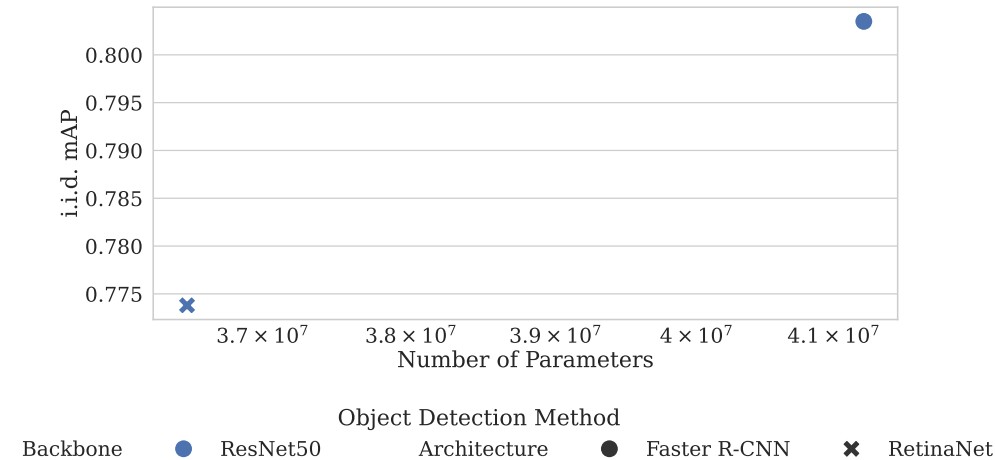

Figure 26: Using the PASCAL VOC dataset for object detection, we benchmark the i.i.d. performance of the available object detection methods. The colors represent the backbone of the respective method, while different marker shapes represent the architecture of the method. The methods were trained on the train set from the PASCAL VOC object detection dataset. The y-axis shows the number of learnable parameters used by the method, and the x-axis shows the mAP performance on i.i.d. samples.

Here, the 'model_folder' contains the configuration and weights for the model for a given dataset. Please note, the 'threat model' is a 3D common corruption type, for example, here 'cc3d_near_focus'. To use DETECBENCH to perform evaluations on all corruptions under the respective 'threat model', use 'all' as the '*name*'. We additionally provide two types of logging, either using 'logger' and/or 'wandb'.

DETECBENCH supports the following 3D Common Corruption: 'color_quant', 'far_focus', 'fog_3d', 'iso_noise', 'low_light', 'near_focus', 'xy_motion_blur', and 'z_motion_blur'. For the evaluation, DETECBENCH will evaluate the model on the validation images from the respective dataset corrupted using each of the aforementioned corruptions for the given severity.

## O    ADDITIONAL RESULTS

Following we include additional results from the benchmark made using DETECBENCH.

### O.1    PASCAL VOC RESULTS

Unfortunately, we could find only two methods trained on the PASCAL VOC dataset for object detection. Nonetheless, we benchmark their performance here. In Figure 26 we report the performance of the object detection methods on the i.i.d. samples. In Figure 27 we report the performance of the object detection methods against all the 2D Common Corruptions and 3D Common Corruptions as severity=3.

### O.2    ALL MS-COCO RESULTS

#### O.2.1    FGSM ATTACK

In Figure 28, we report the evaluations using FGSM attack, both as a non-targeted attack optimized under the $\ell_\infty$-norm bound with a perturbation budget $\epsilon = \frac{8}{255}$.

#### O.2.2    ITERATIVE ATTACKS

In Figure 29, we report the evaluations using BIM and PGD attacks, as non-targeted attacks optimized under the $\ell_\infty$-norm bound with perturbation budget $\epsilon = \frac{8}{255}$, and step size $\alpha$=0.01, over

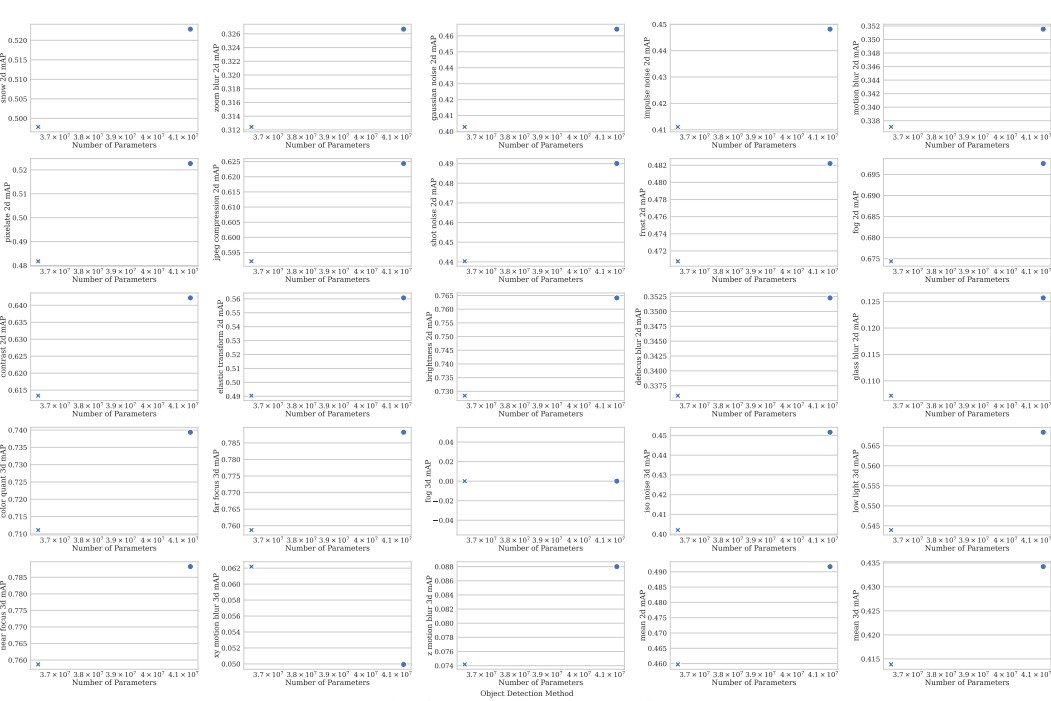

Figure 27: Using the PASCAL VOC dataset for object detection, we benchmark the OOD performance of the available object detection methods against all the 2D Common Corruptions and 3D Common Corruptions. The methods were trained on the train set from the PASCAL VOC object detection dataset. The colors represent the backbone of the respective method, while different marker shapes represent the architecture of the method. The y-axis shows the number of learnable parameters used by the method, and the x-axis shows the mAP performance against the respective common corruption.

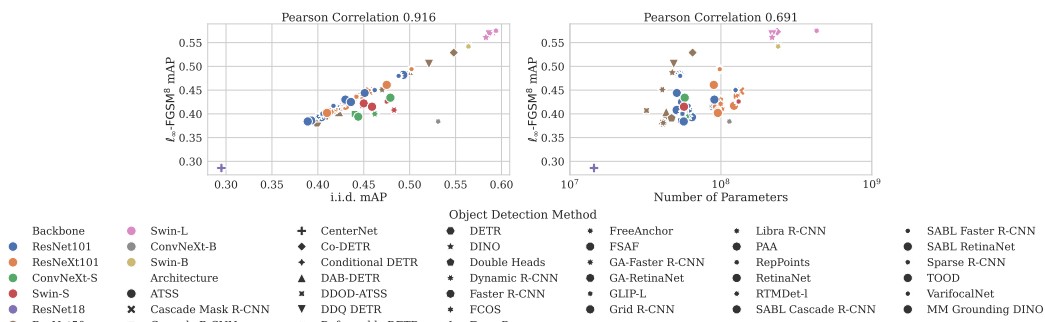

Figure 28: Using the MS-COCO dataset, left: correlation between the performance against FGSM attack and the performance on i.i.d. samples; right: correlation between the performance against FGSM attack and the number of learnable parameters in a method. The colors represent the backbone of the respective method, while different marker shapes represent the architecture of the method. All methods were trained on the train set of the MS-COCO dataset. For most of the methods, we observe a high positive correlation between the i.i.d. performance and the performance against the FGSM attack. However, we observe no correlation between the performance against FGSM attack and the number of learnable parameters in a method.

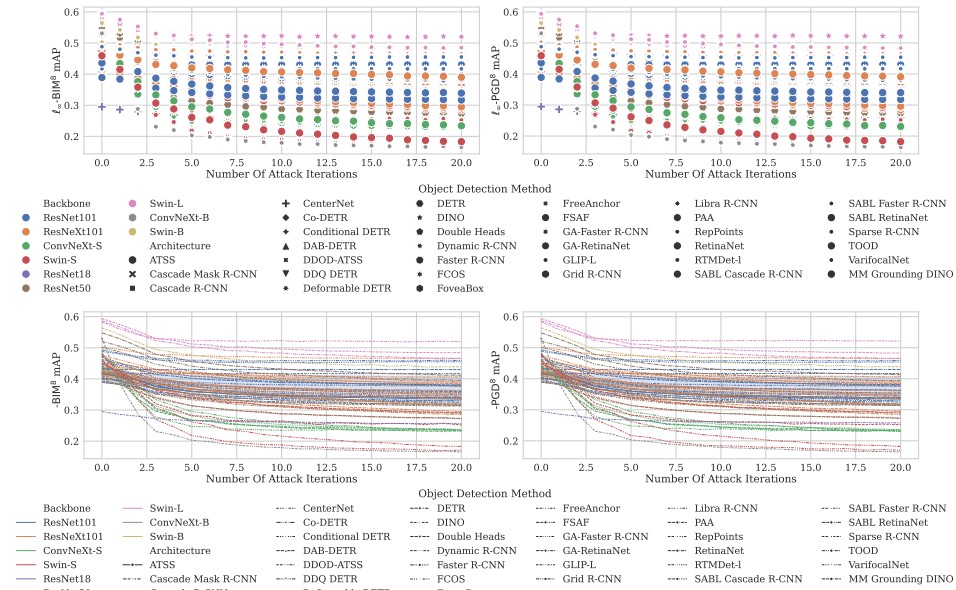

Figure 29: Using the MS-COCO dataset, here we report the mAP performance of all the object detection methods considered in the benchmark against the BIM and PGD adversarial attacks The colors represent the backbone of the respective method, while different marker shapes represent the architecture of the method. The x-axis shows the number of attack iterations used for optimizing the attack, while the y-axis shows the mAP performance. The methods were trained on the training set of the MS-COCO dataset. For ease of understanding, we report this in two ways: the Top is a scatter plot, while the Bottom is a line plot.

attack iterations from 0 to 20, such that at iterations=0, no attack is used i.e. for iterations=0 we report the i.i.d. performance.

O.2.3   2D COMMON CORRUPTIONS

In Figure 30, we report evaluations using the different 2D common corruptions at severity=3 over all the considered methods.

O.2.4   3D COMMON CORRUPTIONS

In Figure 31, we report evaluations using different considered 3D common corruptions at severity=3 over all the considered methods.

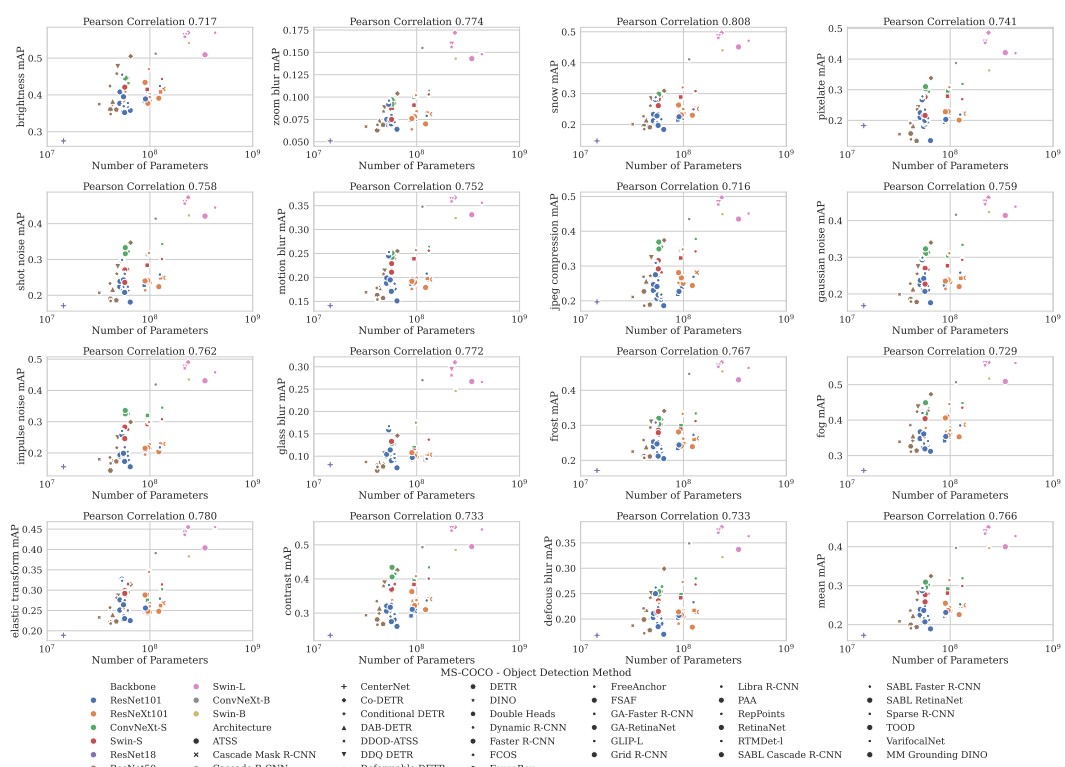

Figure 30: Using the MS-COCO dataset, we report the mAP performance of all considered object detection methods against each of the 2D Common Corruptions evaluated at severity=3. The colors represent the backbone of the respective method, while different marker shapes represent the architecture of the method. The y-axis shows the number of learnable parameters in the method, and the x-axis shows the mAP performance against the respective 2D Common Corruption. All methods were trained on the training set from the MS-COCO dataset.

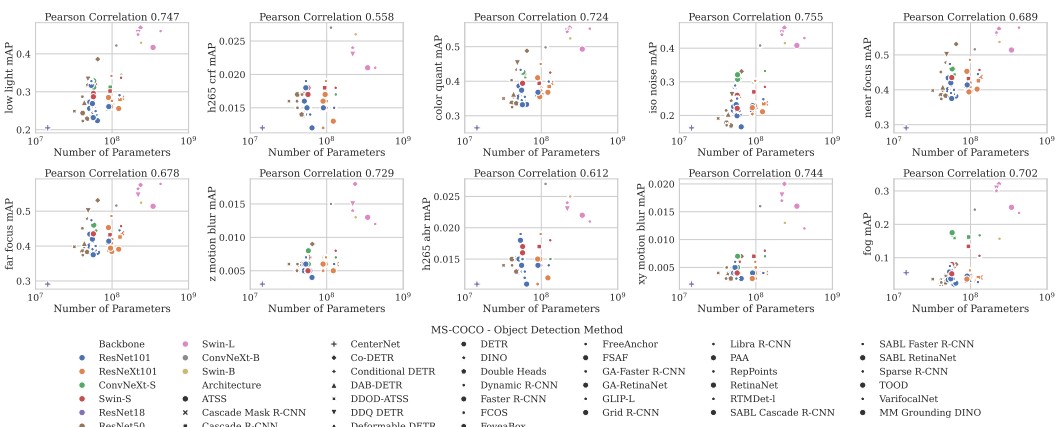

Figure 31: Using the MS-COCO dataset, we report the mAP performance of all considered object detection methods against each of the 3D Common Corruptions evaluated at severity=3. The colors represent the backbone of the respective method, while different marker shapes represent the architecture of the method. The y-axis shows the number of learnable parameters in the method, and the x-axis shows the mAP performance against the respective 3D Common Corruption. All methods were trained on the training set from the MS-COCO dataset.

