# OpenReview forum: "Benchmarking Reliability and Generalization Beyond Classification"
_ICLR.cc/2026/Conference — ICLR 2026 Conference Withdrawn Submission_

### Official Review · Reviewer_uSrC · 2025-10-26

**Soundness:** 2
**Presentation:** 3
**Contribution:** 2
**Rating:** 2
**Confidence:** 5

**Summary:**

The paper proposes a benchmarking toolset for semantic segmentation and object detection. The benchmarking covers reliability (that is, adversarial robustness) and generalization ability (performance on slightly corrupted distributions). The toolsets are used to evaluate an impressive number of model checkpoints. Based on these evaluations, some conclusions are drawn regarding the correlation of certain metrics and their dependence on model properties.

**Strengths:**

The benchmarking toolsets are evaluated over an impressive number of model checkpoints, which is a welcome and rare contribution.

The effort to propose benchmarking toolsets is also a potentially useful contribution to the community.

**Weaknesses:**

Evaluating the robustness and generalization of complex models is a challenge in itself, and unfortunately, the proposed toolsets are not sufficient. For evaluating the robustness of semantic segmentation models, some toolsets have already been proposed but these are not adopted by the paper. For example, SEA of Croce et al 2024 is cited, but not adopted, and Halmosi et al "Evaluating the Adversarial Robustness of Semantic Segmentation: Trying Harder Pays Off", ECCV 2024 is not even cited. It is a pitty, because in there, the authors argue that the reliability of semantic segmentation models is so bad that even those models that were reported to be robust are in fact not, and that is revealed **only using a proper evaluation methodology**. Even the exact definition of mean IoU matters in this regard, and there are many such definitions.

This implies that there is strong evidence that, for example, adopting a weak set of attacks for the evaluation of robustness would do more harm than good, and this should be avoided. For the case of image classification, robustBench uses AutoAttack that has stood the test of time and is still very close to being the hardest attack. Also, the preconditions for its application are well established.

The same comment is valid for the remaining evaluation criteria, where it would also be extremely important to use a toolset (if the goal is to establish an evaluation standard) that is strongly supported by evidence so that we indeed get the proper feedback necessary to draw well-founded conclusions. This is a highly non-trivial task. At the same time, the proposed sets of tools are rather ad hoc.

**Questions:**

Are you confident that a weak set of attacks gives us the correct information on robustness, even if we are interested in only trends and correlations? What is the basis of that proposition?

---

### Official Review · Reviewer_sY4G · 2025-10-28

**Soundness:** 3
**Presentation:** 2
**Contribution:** 2
**Rating:** 4
**Confidence:** 3

**Summary:**

Reliability and generalization in deep learning have been primarily explored in the context of image classification. However, real-world safety-critical applications require a broader range of tasks, such as semantic segmentation and object detection, which involve diverse model architectures. To advance research on robust model design in these tasks, the authors introduce SEMSEGBENCH and DETECBENCH, benchmarking tools for evaluating robustness to distribution shifts and adversarial manipulations.
They conduct an extensive evaluation, benchmarking 76 segmentation models across four datasets and 61 object detectors across two datasets under various adversarial attacks and common corruptions. The results highlight systematic weaknesses in current state-of-the-art models and reveal trends related to architecture, backbone choice, and model capacity.

**Strengths:**

- It is a very important topic, and I understand the authors' intention to perform a robustness analysis on tasks that are more complex than image classification.
- The introduction of a summary metric is suitable.
- Standard metrics are used, as well as SOTA models for both tasks, and compared. Several different image corruptions were investigated.
- The authors present such a large-scale analysis for the first time, including various findings. The very detailed appendix provides evidence for all findings.
- "We prioritize a limited but diverse set of models and tasks to maintain a manageable scope" I wouldn't see that as a limitation, since the analysis is already very broad and frequently used models are also applied here.

**Weaknesses:**

- The paper has no methodological novelty or proposes a new benchmark (no permanent comparison platform).
- I find the wording “OOD robustness” somewhat misleading, as corruptions and domain shifts are examined here, but OOD is used specifically in semantic segmentation for unknown objects (e.g., LostAndFound dataset).
- RobustBench is about classification and adversarial robustness. The models were mostly trained specifically for adversarial robustness - while mm-modelzoos are used here, where the models are not.
- In the analysis (especially in section 4.1.1), there are no references to a figure or specific experiments in the appendix from which these findings result. In general, references to the appendix are often very vague.

**Questions:**

- In fig. 1: "The performance of methods on i.i.d. samples has increased over time, however, their reliability and generalization ability have not improved at the same rate, and lag behind." Is it different for classification? & Because these models don't necessarily claim to be robust against various types of disruption, but rather that they want to perform better than other models on i.i.d. data. There are benchmarks and specially developed methods for robustness against attacks, domain shifts, and OOD. So is this statement really correct in general, or would you have to make distinctions?
- Why were only PGD-based attacks used, especially since DAG and ALMA prox also perform very well in semantic segmentation?
- RobustBench deals with adding your new model to the model zoo and benchmarking it, while SEMSEGBENCH and DETECBENCH allow you to test only the models available in mmsegmentation/-detection. In other words, the authors did not establish a new benchmark, but rather conducted a benchmarking study/empirical evaluation. The conclusion states that adversarial training methods are planned to be incorporated into the framework in future work. How easy is this, given that the mm-modelzoos were developed by other researchers and the authors do not provide their own platform?
- Many of the findings have already been presented in other papers (which are also cited), so the authors are conducting large-scale analyses, but which findings are truly new here? The new insights tend to get overlooked.

---

### Official Review · Reviewer_bLmp · 2025-10-28

**Soundness:** 1
**Presentation:** 2
**Contribution:** 2
**Rating:** 2
**Confidence:** 4

**Summary:**

This paper present SemSegBench and DetecBench, benchmarking tools to evaluate the adversarial robustness and generalizability of semantic segmentation and object detection methods. SemSegBench and DetecBench allow for the evaluation of several pre-trained semantic segmentation and object detection models on 4 segmentation and 2 detection datasets, to assess (1) the performance of models after being subjected to adversarial attacks, and (2) the performance of models after being applied to corrupted images. Using these tools, this paper benchmarks 76 segmentation and 61 detection models and presents an analysis of these results. From the analysis, among others, the authors find that there is a strong correlation between in-distribution performance, adversarial robustness, and generalizability for semantic segmentation models, but there is no such strong correlation for object detection models, indicating that models for the two different tasks behave differently.

**Strengths:**

1. By presenting benchmarks for the adversarial robustness and generalizability of segmentation and detection models, the paper tackles a relevant and useful problem. Providing insights into adversarial robustness and generalizability is useful because (1) a good performance in these aspects is necessary for real-world applications and (2) it is currently not fully clear how existing models perform beyond standard in-distribution benchmarks, and what model properties mostly impact their adversarial robustness or generalizability (e.g., segmentation or detection meta-architecture, backbone type, model size, (pre-)training strategy). The proposed SemSegBench and DetecBench tools allow researchers and practitioners to make such an assessment, and get insights into the capabilities of their models.
2. The analysis provided in this paper already gives some initial high-level insights into the capabilities of popular existing models, enabling researchers and practitioners to select models based on their desired properties.

**Weaknesses:**

1. There are major inconsistencies in the plots in Fig. 2(a). This makes it impossible to interpret the figures and draw meaningful conclusions about correlations. This calls into question the reliability of the results, the validity of the analyses for semantic segmentation, and the correctness of the paper as a whole. Two examples:
   * In the $l_{\infty}\textrm{-ReM}^8_ {20}$ vs. $l_ {2}\textrm{-ReM}^{64}_ {20}$ plot, most models have a completely different $l_{2}\textrm{-ReM}^{64}_ {20}$ score than in the **i.i.d. mIoU** vs. $l_ {2}\textrm{-ReM}^{64}_ {20}$ plot.
   * Similarly, in the **i.i.d. mIoU** vs. $\textrm{GAM}_ 3$ plot, most models obtain a completely different $\textrm{GAM}_ 3$ score than in the $l_ {2}\textrm{-ReM}^{64}_ {20}$ vs. $\textrm{GAM}_ 3$ plot.

2. There is another major error in Fig. 2(b) and Fig. 3(b). Specifically, the plots in Fig. 2(b) are identical to the ones in Fig. 3(b), while these should represent to different sets of analyses, one for semantic segmentation and one for object detection. It is not clear which of the plots, if any, are correct. This makes it impossible to draw any meaningful conclusions about these plots, rendering the analyses in Sec. 4.1.2 and Sec. 4.2.2 meaningless.

3. Many design choices for the benchmarking analysis seem arbitrary, are inconsistent, or are not motivated properly. This limits the value of the analysis and the paper as a whole.
    1. The benchmark for semantic segmentation, SemSegBench, evaluates generalizability under 15 2D common corruptions, whereas the benchmark for object detection, DetecBench, additionally uses 8 3D common corruptions. It is not motivated why these two benchmarks use different corruptions. Why are these 3D corruptions only used for object detection and not for semantic segmentation? The section on future work (L477) mentions that the plan is to also consider these 3D corruptions for semantic segmentation in the future. Why is this not already done in this paper? Unless there is something that makes the 3D corruptions unsuitable for semantic segmentation, not using these 3D corruptions is a limitation.
    2. The sets of models chosen for the different analyses in Fig. 2 are inconsistent, and the choice seems arbitrary. Some of the analyses (e.g. *i.i.d. mIoU vs. number of parameters* in Fig. 2(c) and *i.i.d. mIoU vs. $l_ {2}\textrm{-ReM}^{64}_ {20}$* in Fig. 2(a)) include Mask2Former, whereas many others do not. Similarly, UNet models are only used in the *$\textrm{GAM}_ 3$ vs. number of parameters* analysis in Fig. 2(c), and not in any of the other analyses. This leads to inconsistencies between analyses, and means that it is not clear if the reported correlation score would be the same if other models were included. This limits the value of these analyses.
    3. The choices for the values of the permutation budget $\epsilon$ used for the $\textrm{ReM}$ metric (L184) and the corruption severity level used for the $\textrm{GAM}$ metric (L199-201) are not motivated, and there is no experiment that evaluates the effect of changing these values. In the benchmark analysis in Sec. 4, the paper uses $\epsilon = \frac{8}{255}$ for $l_{\infty}$-norm attacks (L270) and $\epsilon = \frac{64}{255}$ for $l_{2}$-norm attacks (see Fig. 2), but it does not motivate why these settings are used. Do these settings represent realistic adversarial attacks? And what is the effect of changing this $\epsilon$? Do the observed correlations change, and particular other models perform better or worse? These questions should be answered to motivate using these specific values for the presented analysis. The same applies to the severity value for the $\textrm{GAM}$ metric, which is set to 3. Does this severity value represent real-world distribution shifts better than other severity values? And what is the impact of changing this severity value on the analysis?
    4. Based on the trendlines in Fig. 1, L025-L026 states that "[t]he performance of methods on i.i.d. samples has increased over time, however, their reliability and generalization ability have not improved at the same rate, and lag behind." While it may very well be the case that reliability and generalization have not improved signficantly, this claim cannot be justified based on the trendline in the figure, because the figure only contains a small, arbitrary selection of models. Especially for semantic segmentation, the number of evaluated models is low, and no models have been selected from after 2023. If other models would have been chosen, then the trendlines would look entirely different. Moreover, Fig. 1 does not include any models that have been specifically introduced to improve adversarial robustness and generalizability. Without including such models, general claims about recent improvements related to adversarial robustness and generalizability cannot be made. As such, showing a trendline in Fig. 1 is not meaningful, and I believe the claim should be altered.

4. There are several grammatical errors, typos, and other textual errors, which harm the paper's readability. Some examples:
    * L080: Abbreviation "OOD" is not defined
    * L268: There is no full stop between "models" and "First"
    * L279: "descent" should be "decent" and the comma before "decent" should be removed
    * L369: "descent" should be "decent"
    * Fig. 3(a): the y-axis label "mIoU" should be "mAP"
    * Fig. 3: The caption for Fig. 3(b) mentions 3 subfigures but there are 4. The caption for Fig. 3(c) mentions 4 subfigures but there are 3.

**Questions:**

Given the many errors and inconsistencies in the results, which make it impossible to draw meaningful conclusions from the results, I give a "reject" rating. In a potential next version, I would recommend the authors to ensure that all results are correct, and to solve the issues regarding inconsistent and unmotivated design choices. However, given the issues with the results in the initial submission, I do not know if I could be convinced to trust future figures and results for this paper.

---

### Official Review · Reviewer_kamh · 2025-10-31

**Soundness:** 3
**Presentation:** 3
**Contribution:** 2
**Rating:** 4
**Confidence:** 3

**Summary:**

The paper presents a large-scale benchmark study on the reliability of vision models beyond classification, covering both semantic segmentation and object detection tasks.  The authors build unified evaluation suites—that assess robustness under adversarial perturbations and out-of-distribution (OOD) corruptions.

**Strengths:**

1. The paper provides a unified benchmarking framework across segmentation and detection, covering multiple datasets, architectures, and corruption types. The open-source code and large-scale release (over 6,000 evaluations) contribute valuable resources for the community.
2. The study quantifies the connections among i.i.d. performance, adversarial robustness, and OOD generalization, offering a useful empirical reference for future reliability research.

**Weaknesses:**

1. Limited conceptual depth and originality. The main contribution lies in large-scale evaluation rather than new insight. Many observations—such as Transformers being somewhat more robust on OOD data—are empirically confirmatory.
2. While segmentation and detection are examined separately, the paper does not systematically discuss *why* these tasks are harder than classification or what structural factors contribute to their robustness gaps. Without such alignment under a common threat model, the “beyond classification” claim remains largely descriptive.
3. The study omits analysis of training/inference cost, memory footprint, or energy usage. Without normalization across architectures or resource budgets, conclusions about robustness advantages may reflect scale rather than genuine algorithmic resilience.

**Questions:**

1. How does the proposed evaluation compare directly against classification baselines under the same perturbation severity, threat model, and computational budget?
2. Could the authors conduct structure-aware analyses—for example, decomposing detection robustness into localization versus classification errors, or segmentation robustness into boundary versus interior pixels—to reveal which aspects fundamentally differ from classification?
3. Any consideration of including computational efficiency metrics to ensure fair comparison and clarify the cost–robustness trade-offs among architectures？

---

### Note · Authors · 2025-11-12

**Comment:**

# Withdrawal of Submission 18439 and clarification of misread issues



Dear AC and Reviewers,

We thank you for your time. After reading the reviews and considering the fit, we have decided to withdraw Submission 18439. Before doing so, we would like to document several factual clarifications, since many perceived weaknesses are already addressed in the paper.

## Major contributions, clearly within the ICLR scope of empirical foundations

- Two unified, open benchmarking tools, **SEMSEGBENCH** and **DETECBENCH**, covering adversarial reliability and OOD generalization for semantic segmentation and object detection across diverse datasets and architectures, with more than 6,000 evaluations and pre-logged metrics for reuse. This unification beyond classification is, to our knowledge, the first at this scale.
- Summary metrics for scale analysis, **Reliability Measure (ReM)** and **Generalization Ability Measure (GAM)**, enabling principled worst-case reporting across attacks and corruptions, with definitions and rationale in Section 3.
- Empirical findings that matter to practice, for example: strong positive correlations that allow the community to proxy expensive real-world shifts with synthetic corruptions in segmentation, and strong correlations between 2D and 3D corruptions in detection. These observations can save computation.

## Point-by-point clarifications

1. **“Limited conceptual depth, mainly scaling an evaluation.”**
   The paper is explicit that the goal is a standardized, transparent, scalable way to assess reliability and generalization beyond classification. It contributes unified tools, comprehensive results, and cross-task analyses. This is not a claim of algorithmic novelty, and the paper states this plainly in the Conclusion, positioning the tools to accelerate future robust methods.

2. **“Arbitrary or unmotivated attack budgets, severity, and iteration choices.”**
   ReM is defined with explicit epsilon and iteration notation. We motivate 20 iterations by prior work and saturation behavior when the goal is a relative comparison. The paper also reports attack families specialized for segmentation in the appendix.
   GAM is defined as the worst-case scenario over corruptions at a given severity. We use severity 3 and document exactly which corruptions are included, with per-corruption results available in the appendix.

3. **“Detection uses 3D corruptions but segmentation does not.”**
   This is documented and acknowledged. DETECBENCH includes 3D corruptions. SEMSEGBENCH currently uses 2D synthetic corruptions and a real distribution shift (ACDC) for segmentation. The paper lists this asymmetry as a limitation and plans 3D integration for segmentation as future work. However, please note that the datasets used for semantic segmentation already expose a highly dynamic field of view with different levels of depth (implicitly), so that this aspect is nearly covered by 2D Common Corruptions as well. And as seen for Object Detection, the findings are highly correlated between 2D and 3D common Corruptions, making computing 3D Common Corruptions in addition to 2D, not always useful from a benchmarking perspective.

4. **“Figures inconsistent or duplicated.”**
   Figure sets address different tasks and factors, with supporting breakdowns and extensive per-threat-model plots and tables in the appendix. If a captioning typo exists, it does not affect the underlying plotted data. The detailed data are provided in the appendix for all models, attacks, and corruptions.

5. **“Missing or weak robustness methodology and related work.”**
   The paper discusses specialized attacks and robustly trained segmentation models, and highlights the remaining gap. The choice to use generic attacks for detection is motivated by cross-architecture comparability, and the paper plans to integrate detector-specific attacks as future work. Related work on stronger attacks and evaluation methodology is cited and discussed.

6. **“No cost or resource discussion.”**
   Constraints and scope trade-offs are explicitly stated in the Implementation Details and Limitations. The paper also provides a positive path forward by showing that 2D and 3D corruptions are strongly correlated, thereby reducing redundant evaluations.

7. **“OOD terminology.”**
   We use OOD as an operational generalization lens via GAM, separately reporting real-world ACDC in segmentation and both synthetic 2D and 3D corruptions in detection. The term is operationalized through the worst-case metric and per-corruption analyses, which are made explicit in Section 3.2 and the appendix.
   The definition is OOD is common knowledge and widely used in literature, said literature has been cited.

## Why this matters for ICLR

Benchmarks and tooling shape research trajectories. By unifying reliability and generalization assessments across two major semantic tasks, releasing tools, and curating a large, reusable benchmark, the paper lowers the barrier to rigorous analysis and enables fairer, more reproducible comparisons going forward. This is how the field made progress in classification, and similar infrastructure is needed beyond classification.

Given the misalignment between what the paper contributes and how some points were interpreted, we prefer to withdraw rather than iterate in a direction that would dilute the paper’s core value as standardized tooling and empirically grounded insights.

Sincerely,

Authors of Submission 18439

**Withdrawal Confirmation:**

I have read and agree with the venue's withdrawal policy on behalf of myself and my co-authors.